# CHRONOSCORE: CONTEXT-AWARE SCHEDULING VIA SLACK-DRIVEN TEMPORAL REASONING

## ABSTRACT

Real-time deployments require schedulers that reason about complex temporal interactions while meeting strict latency budgets. We present **ChronosCore**, a value-based scheduler that embeds a compact, permutation-invariant Transformer inside a Deep Q-Network and represents per-job urgency through the *Urgency Tokenizer*, a learnable slack-quantization layer. These discrete urgency tokens feed an attention module that models cross-task dependencies with low computational cost. The design integrates latency-aware sparse attention and practical multi-core mapping, enabling global reasoning under tight inference constraints. Empirical studies across single-core benchmarks, industrial mixed-criticality traces, and large multiprocessor workloads show consistent gains in deadline adherence and responsiveness over classical and learned baselines. Complementary analyses including encoder and quantization ablations, guidance for selecting discretization parameters, attention-based interpretability, and hardware micro-benchmarks clarify when and why the approach is effective. ChronosCore offers a practical path for deploying attention-driven decision policies in latency-sensitive scheduling systems.

**Keywords:** Reinforcement Learning, Real-Time Systems, Transformer Models, Attention Mechanisms, Scheduling under Uncertainty, Resource Allocation, Embedded AI

## 1 INTRODUCTION

Real-time systems require schedulers that make correct, low-latency decisions under dynamic workloads. Classical policies such as Rate Monotonic and Earliest Deadline First provide strong guarantees under ideal assumptions but degrade under bursty loads or uncertain execution times, motivating multicore strategies and empirical studies beyond idealized conditions (Phan et al., 2011; Abeni & Cucinotta, 2020).

Data-driven approaches address these limitations by learning policies from interaction data. RL has shown promise for cloud orchestration, job-shop scheduling, and cluster placement (Wang et al., 2024a; Cheng et al., 2022; Zhang et al., 2022; Lei et al., 2023; Li et al., 2023a), with offline RL and imitation learning improving sample efficiency in constrained domains (Remmerden et al., 2025). However, many RL schedulers rely on sequence encodings or fixed-size vectors, introducing order dependence and limiting generalization (Jalali Khalil Abadi et al., 2024; Swarup et al., 2021). Set- and graph-based models mitigate these issues, yet integrating them under strict sub-millisecond inference budgets remains challenging (Li et al., 2024a). Transformers enable global reasoning via attention, making them attractive for scheduling where cross-task interactions matter. Multi-head attention supports parallel pairwise modeling (Vaswani et al., 2017; Cao et al., 2024; Chen et al., 2021c). Sequence-modeling approaches such as Decision Transformer excel in offline RL but depend on ordered histories and causal masking, unsuitable for unordered sets and tight latency constraints (Chen et al., 2021a). Online attention-based agents improve representation capacity (Hu et al., 2024), while hardware co-designs enhance throughput without guaranteeing tail-latency bounds (Moon et al., 2025). Sparse and selective attention methods, including explicit key selection and block-sparse routing, offer compression strategies, and RL-guided quantization reduces runtime cost; however, adapting these techniques to hard real-time value-based schedulers requires careful co-design of representation, sparsification, and mapping strategies (Zhao et al., 2019; Lou et al., 2024; Kwon et al., 2024; Roy et al., 2021; Zhou et al., 2025) We introduce ChronosCore,

a value-based RL scheduler designed for predictable, low-latency operation with global reasoning. ChronosCore combines three design choices: a slack-quantized token representation that discretizes continuous slack into learnable embeddings, reducing gradient variance and focusing attention on deadline-aware groups; a compact, permutation-invariant attention encoder with shallow depth, narrow width, and sparsification via block Top-$k$ and locality-aware chunking for near-linear scaling; and multicore mapping layers that translate per-token Q-values into core assignments under latency and migration constraints using masked-greedy or bipartite matching variants. ChronosCore is trained with stable value-based updates and engineered exploration schedules for robustness under overload. Experiments on uniprocessor, mixed-criticality, and large-scale multiprocessor workloads show consistent gains in deadline compliance and response time over analytic and learned baselines, while maintaining sub-millisecond inference. Additional analyses include quantization and encoder ablations, attention DQN interpretability, and tail-latency micro-benchmarks across hardware targets.

In summary, our contributions are threefold. We introduce the Urgency Tokenizer, a principled slack-quantization module that provides a compact deadline-aware embedding and stabilizes value learning. We develop a lightweight, permutation-invariant Transformer Q-network with a latency-aware sparsification scheme that enables global reasoning at low cost. We integrate practical multi-core mapping with learned Q-scores and validate the full system on synthetic and industrial traces with comprehensive diagnostics and guidance for selecting quantization parameters.

## 2 RELATED WORK

### 2.1 CLASSICAL REAL-TIME SCHEDULING

Priority-based policies such as Rate Monotonic and Earliest Deadline First provide schedulability guarantees under ideal assumptions, with RM assigning static priorities by period and EDF achieving optimality on a single preemptive processor. These guarantees degrade under overload or uncertain execution times, motivating alternative frameworks and multicore strategies such as Pfair and LLREF, along with empirical studies beyond idealized conditions (Abeni & Cucinotta, 2020; Phan et al., 2011).

### 2.2 LEARNING-BASED AND RL SCHEDULERS

RL-based scheduling has been applied across cloud, edge, manufacturing, and cluster domains using latency-aware DQN for orchestration (Wang et al., 2024a; Cheng et al., 2022), PPO and hierarchical RL for job-shop tasks (Zhang et al., 2022; Lei et al., 2023), and graph-structured or multi-agent models for large-scale placement (Zhao & Wu, 2021; Fan et al., 2022). Recent work addresses parallel-machine and manufacturing problems with transformer-enhanced RL (Li et al., 2023a). Offline RL and imitation learning improve sample efficiency via historical traces (Remmerden et al., 2025; Yang et al., 2025). A recurring limitation is reliance on sequence encodings or hand-crafted features, which hinder permutation-invariant generalization; empirical studies highlight these issues (Jalali Khalil Abadi et al., 2024; Swarup et al., 2021). Set- and graph-based architectures mitigate ordering constraints, but integrating them into value-based RL under strict latency budgets remains challenging (Chen et al., 2021b; Li et al., 2024a).

### 2.3 TRANSFORMER-BASED RL AND EXPLICIT COMPARISONS

Transformer-based RL splits into offline sequence-modeling such as Decision Transformer for offline RL (Chen et al., 2021a) and online agents that incorporate attention for richer representations while retaining bootstrapping and value estimation (Hu et al., 2024). Multi-head attention enables parallel pairwise reasoning and long-range dependency modeling (Vaswani et al., 2017; Cao et al., 2024; Chen et al., 2021c). Trajectory transformers, however, require ordered histories and causal masking, conflicting with permutation invariance for unordered task sets, and are trained offline with supervised objectives, whereas real-time scheduling demands low-latency, on-policy updates. Heavy transformer deployments and hardware accelerators prioritize throughput rather than strict tail-latency guarantees (Chen et al., 2021a; Moon et al., 2025; Hu et al., 2024). These differences make Decision Transformer–style methods unsuitable for predictable sub-millisecond scheduling workloads.

### 2.4 TRANSFORMERS, SPARSE ATTENTION AND EFFICIENT ARCHITECTURES

Dense self-attention scales quadratically with token count, making it costly for large task sets. Efficiency can be improved through salient-key selection and concentrated attention (Zhao et al., 2019), algorithmic sparse schemes that trade minor accuracy loss for runtime gains (Lou et al., 2024), and system-level strategies such as RL-guided mixed-precision and hardware acceleration (Kwon et al., 2024; Moon et al., 2025). Additional work on explicit sparse selection, routing, block-sparse techniques, and transformer co-design informs practical compression strategies (Roy et al., 2021; Zhou et al., 2025; Gao et al., 2025; Yue et al., 2024; Gupta et al., 2025). These approaches collectively motivate the sparsification and chunking recipes we adopt to balance global reasoning with strict latency budgets.

### 2.5 WHERE CHRONOSCORE STANDS

ChronosCore integrates scheduling theory, reinforcement learning, and efficient transformer design to deliver predictable low-latency operation within a value-based RL loop. It employs an attention encoder for unordered sets and slack-quantized embedding for compact timing representation, unlike trajectory transformers that depend on ordered histories or large contexts. Compared to heavy transformer or GNN-based dispatch models, ChronosCore prioritizes a small footprint, explicit multi-core action mapping, and empirical micro-benchmarks for decision quality and real-time performance, enabling global reasoning under strict latency constraints (Li et al., 2024a; Chen et al., 2021b; Hu et al., 2024).

## 3 METHODOLOGY

We model real-time scheduling as an MDP and introduce ChronosCore, a value-based agent combining a compact Transformer encoder with the pluggable Urgency Tokenizer (UT). The design covers the task model, UT, UT-enabled training/inference loop, encoder and projection, multicore mapping, learning objective, and interpretability diagnostics.

### 3.1 CHRONOSCORE ARCHITECTURE

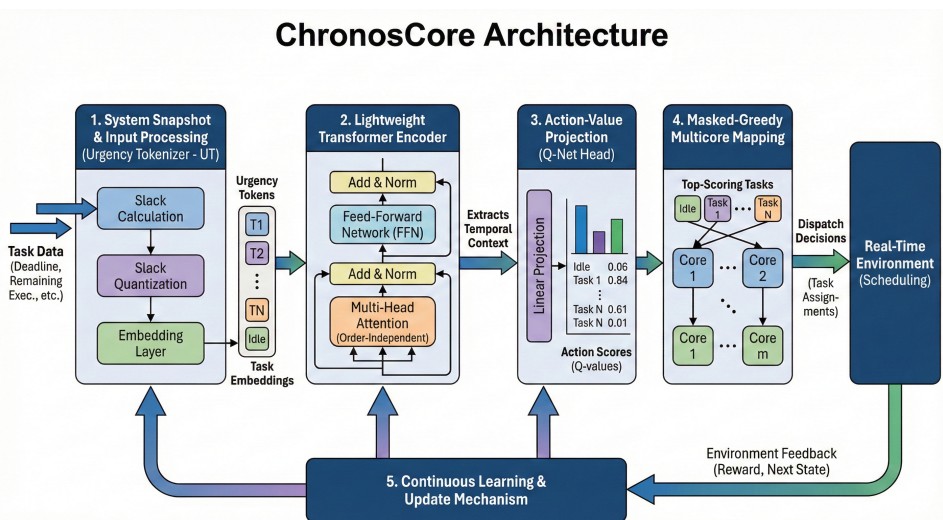

Figure 1: ChronosCore architecture. ChronosCore performs real-time scheduling through a unified pipeline that converts system snapshots into dispatch decisions. It first applies the Urgency Tokenizer to convert continuous slack into learned discrete embeddings, then processes all job tokens with a lightweight, permutation-invariant Transformer using latency-aware sparse attention. The resulting representations are projected into action values and mapped to multicore assignments via masked-greedy selection. A Deep Q-Learning loop continuously refines the policy from environment feedback, improving deadline compliance and responsiveness.

## 3.2 PROBLEM FORMULATION

Consider a task set $\mathcal{T} = \{T_i\}_{i=1}^N$, where each task $T_i$ is described by $(\mathrm{id}_i, P_i, C_i, D_i)$. Time is discrete and indexed by $t \in \mathbb{N}_0$. The $k$-th job instance of task $i$ has release time $r_i^{(k)}$ and absolute deadline $d_i^{(k)} = r_i^{(k)} + D_i$.

$$\mathrm{id}_i \in \{1, \ldots, N\}, \quad P_i > 0, \quad C_i > 0, \quad D_i > 0. \tag{1}$$

where $P_i$ denotes the nominal period, $C_i$ the worst-case execution time and $D_i$ the relative deadline.

Let $c_i(t) \in \{0, 1, \ldots, C_i\}$ be the remaining execution of the active job of task $i$ at time $t$. The uniprocessor action space is

$$a_t \in \mathcal{A} = \{idle, 1, \ldots, N\}, \tag{2}$$

where an integer action selects the corresponding task for execution and 'idle' dispatches none. For $m$ identical cores the per-step decision assigns up to $m$ distinct tasks or idles.

The per-step reward balances completions and deadline misses:

$$r(t) = \sum_{i=1}^N \Big[ \mathbb{I}\{c_i(t-1) > 0 \wedge c_i(t) = 0\} - \mathbb{I}\{t = d_i^{(k)} \wedge c_i(t) > 0\} \Big], \tag{3}$$

where $\mathbb{I}\{\cdot\}$ is the indicator function, $d_i^{(k)}$ denotes the active job's absolute deadline and $c_i(t)$ its remaining execution at time $t$.

## 3.3 URGENCY TOKENIZER (UT): A PLUGGABLE LEARNABLE QUANTIZATION LAYER

We introduce the *Urgency Tokenizer* (UT), a reusable module that converts continuous per-job slack into a small vocabulary of learned urgency tokens. UT is treated as a first-class layer in the model pipeline. UT performs three steps: discretize slack to an index, look up a trainable embedding, and return the urgency token for downstream encoding.

Per-job slack is defined as

$$s_i(t) = \big(d_i^{(k)} - t\big) - c_i(t), \tag{4}$$

where $d_i^{(k)}$ is the absolute deadline of job instance $k$, $t$ is the current time, and $c_i(t)$ is the remaining execution.

UT maps $s_i(t)$ to a quantized index $\tilde{s}_i(t)$ and an embedding vector $\mathbf{x}_i(t)$:

$$\tilde{s}_i(t) = \mathrm{clip}\Big( \Big\lfloor \frac{s_i(t)}{\Delta} \Big\rfloor, 0, Q - 1 \Big), \tag{5}$$

$$\mathbf{x}_i(t) = \mathbf{E}\big[\tilde{s}_i(t)\big] \in \mathbb{R}^d, \tag{6}$$

where $Q$ is the number of quantization levels, $\Delta > 0$ is the bin width, $\lfloor \cdot \rfloor$ is the floor operator, $\mathrm{clip}(\cdot, 0, Q-1)$ bounds indices to the valid range, $\mathbf{E} \in \mathbb{R}^{Q \times d}$ is a trainable embedding matrix, and $d$ denotes the embedding dimension. The vector $\mathbf{x}_i(t)$ is the urgency token provided to the encoder.

## 3.4 UNIFIED ALGORITHM: UT-ENABLED TRAINING AND ONLINE DECISION

The unified procedure below integrates UT into the main training and online decision loop. Each reference to an equation label below points to the corresponding definition above.

## 3.5 ENCODER, ATTENTION AND POSITIONAL STRATEGY

The encoder consumes urgency tokens (and optional per-job features) and returns contextualized representations. Define the input token matrix as

$$\mathbf{X}(t) = \big[\mathbf{x}_0(t); \mathbf{x}_1(t); \ldots; \mathbf{x}_N(t)\big] \in \mathbb{R}^{(N+1) \times d}, \tag{7}$$

---

**Algorithm 1:** ChronosCore: UT-enabled Training and Online Decision

---

**Input** : Episodes $M$, steps per episode $T$, slack bin width $\Delta$, quantization levels $Q$,
embedding table $\mathbf{E}$, learning rate $\alpha$, target mixing $\tau$, exploration schedule

**Output:** Trained Q-network $Q_\theta$

1 Initialize primary network $Q_\theta$ and target network $Q_{\theta^-} \leftarrow Q_\theta$;

2 Initialize replay buffer $\mathcal{D} \leftarrow \varnothing$;

3 **for** *episode* $\leftarrow 1$ **to** $M$ **do**

4     Reset environment and observe initial state;

5     **for** $t \leftarrow 0$ **to** $T - 1$ **do**

6        **for** *each active job* $i$ **do**

7           compute slack $s_i(t) \leftarrow (d_i^{(k)} - t) - c_i(t)$;       //see Eq. equation 4

8           $q \leftarrow \mathrm{clip}(\lfloor s_i(t)/\Delta \rfloor, 0, Q - 1)$;      //discretize slack; see
           Eq. equation 5

9           $\mathbf{x}_i(t) \leftarrow \mathbf{E}[q]$;     //embedding lookup (urgency token); see
           Eq. equation 6

10        **end**

11        assemble token matrix $\mathbf{X}(t) \leftarrow [\mathbf{x}_0(t); \mathbf{x}_1(t); \ldots; \mathbf{x}_N(t)]$;

12        $\mathbf{q}(t) \leftarrow \textsc{EncoderForward}(\mathbf{X}(t))$; //per-token Q-scores (Transformer
       + projection); see Eq. equation 11

13        map $\mathbf{q}(t)$ to one or more actions using multicore mapping and execute;

14        observe reward $r_t$ and next state $s_{t+1}$; store $(s_t, a_t, r_t, s_{t+1})$ into $\mathcal{D}$;

15        **if** *training condition is satisfied* **then**

16           sample minibatch $\mathcal{B} \sim \mathcal{D}$;

17           compute targets $y$ using Eq. equation 13;           //TD target

18           update $\theta$ by minimizing Eq. equation 14;           //TD loss

19           soft-update target network: $\theta^- \leftarrow \tau\theta + (1 - \tau)\theta^-$;

20        **end**

21     **end**

22 **end**

---

where $\mathbf{x}_0(t)$ is a learned idle token and $\mathbf{x}_i(t)$ are UT embeddings possibly concatenated with normalized remaining execution and task identifiers.

The encoder stacks $L$ Transformer blocks with residual connections and layer normalization. Let $\mathbf{H}^{(0)} = \mathbf{X}$. For $\ell = 1, \ldots, L$:

$$\mathbf{Z}^{(\ell)} = \mathrm{LayerNorm}\big(\mathbf{H}^{(\ell-1)} + \mathrm{MultiHeadAttn}(\mathbf{H}^{(\ell-1)})\big), \tag{8}$$

$$\mathbf{H}^{(\ell)} = \mathrm{LayerNorm}\big(\mathbf{Z}^{(\ell)} + \mathrm{FFN}(\mathbf{Z}^{(\ell)})\big), \tag{9}$$

where FFN denotes the position-wise feed-forward subnetwork. The attention kernel uses scaled dot-products:

$$\mathrm{Attention}(\mathbf{Q}, \mathbf{K}, \mathbf{V}) = \mathrm{softmax}\left(\frac{\mathbf{Q}\mathbf{K}^\top}{\sqrt{d_k}}\right)\mathbf{V}, \tag{10}$$

where $\mathbf{Q}, \mathbf{K}, \mathbf{V}$ are linear projections of the input, $d_k$ is the per-head dimension and $H$ the number of heads. Absolute positional encodings are omitted to preserve permutation invariance over the unordered job set.

To control runtime cost we employ sparsification such as block Top-$k$ pruning and locality-aware chunking; details and ablations appear in the implementation appendix.

### 3.6 ACTION-VALUE PROJECTION AND MULTICORE MAPPING

After the final encoder layer we compute per-token Q-scores by a linear projection:

$$\mathbf{q}(t) = \mathbf{W}_q \mathbf{H}^{(L)}(t)^\top + \mathbf{b}_q \in \mathbb{R}^{N+1}, \tag{11}$$

where $\mathbf{W}_q \in \mathbb{R}^{(N+1) \times d}$ and $\mathbf{b}_q \in \mathbb{R}^{N+1}$ are learnable and indices correspond to $[idle, 1, \ldots, N]$.

For the uniprocessor the chosen action is

$$a_t = \arg \max_{a \in \{idle, 1, \ldots, N\}} \mathbf{q}_a(t). \tag{12}$$

For $m$ cores we use an iterative masked-greedy mapping in the main system: repeatedly select the highest unmasked token and mask it until $m$ tasks are chosen or only idle tokens remain. An alternative uses a bipartite assignment solved by a differentiable matching layer.

### 3.7 LEARNING OBJECTIVE AND OPTIMIZATION

ChronosCore is trained under Deep Q-Learning with experience replay and a soft-updated target network. For a sampled transition $(s, a, r, s')$ the TD target is

$$y = r + \gamma \max_{a'} Q_{\theta^-}(s', a'), \tag{13}$$

where $\gamma$ is the discount factor and $Q_{\theta^-}$ the target network. The loss minimized over minibatches $\mathcal{B}$ is the mean-squared TD error:

$$\mathcal{L}(\theta) = \mathbb{E}_{(s,a,r,s') \sim \mathcal{B}} \big[ (y - Q_\theta(s, a))^2 \big]. \tag{14}$$

Target parameters are updated by Polyak averaging:

$$\theta^- \leftarrow \tau\theta + (1 - \tau)\theta^-, \tag{15}$$

where $\tau \in (0, 1]$ is the mixing coefficient. Exploration uses an $\epsilon$-greedy schedule with linear annealing from $\epsilon_0$ to $\epsilon_{\min}$.

### 3.8 INTERPRETABILITY DIAGNOSTICS

We extract diagnostics from the final-layer attention maps $\mathbf{A}^{(L)}(t) \in \mathbb{R}^{(N+1) \times (N+1)}$. Alignment is defined as

$$\text{Alignment} = \frac{1}{T} \sum_{t=1}^{T} \mathbb{I}\Big[ \arg \max_j \mathbf{A}_{0j}^{(L)}(t) = a_t \Big], \tag{16}$$

where $T$ is the number of decision timesteps, $\mathbf{A}_{0j}^{(L)}(t)$ denotes attention from the decision token (index 0) to token $j$, and $a_t$ the chosen action. Entropy at time $t$ is

$$\text{Entropy}(t) = - \sum_{j=0}^{N} \mathbf{A}_{0j}^{(L)}(t) \log \mathbf{A}_{0j}^{(L)}(t), \tag{17}$$

which measures concentration of attention mass; reported entropy values are averaged across timesteps.

## 4 EXPERIMENTAL EVALUATION

### 4.1 EXPERIMENTAL SETUP

We conduct comprehensive evaluations across three computational scenarios: uniprocessor periodic scheduling with standardized task configurations, industrial multi-core workloads with mixed-criticality workflows, and large-scale multiprocessor systems with 100–600 tasks. Our framework is benchmarked against established scheduling approaches including classical schedulers (RM, EDF), feedforward Deep Q-Network (FF-DQN), Dynamic Importance-aware Online Scheduling (DIOS), and quantum-inspired optimization methods. **Evaluation Metrics:** Performance assessment employs deadline compliance rate, average response time, and computational overhead.

## 4.2 UNIPROCESSOR PERIODIC SCHEDULING PERFORMANCE

### 4.2.1 STANDARD TASK CONFIGURATION ANALYSIS

Classical scheduling theory establishes that Earliest Deadline First (EDF) is optimal for preemptive, independent periodic tasks on a uniprocessor, provided that the total system utilization satisfies $U \leq 1$ (Liu & Layland, 1973). However, when $U > 1$, no scheduling algorithm, including EDF, can guarantee that all deadlines will be met (Baruah & Haritsa, 2002). This limitation is especially critical in real-world systems, where transient overloads frequently occur. To assess performance under such conditions, we conducted a rigorous evaluation using a representative task configuration with temporal attributes: $T_1 = 40$ms (short-period), $T_2 = 60$ms (medium-period), and $T_3 = 100$ms (long-period). ChronosCore achieved a deadline compliance rate of 79.00%, which corresponds to a 7.57% absolute improvement over feedforward DQN implementations and a 67.33% enhancement compared to conventional schedulers such as EDF and RM. These classical methods failed to meet deadlines under overload, whereas ChronosCore maintained robust performance. This result highlights ChronosCore's practical advantage in real-time systems operating near or beyond nominal capacity, where traditional schedulers are no longer effective.

Table 1: Deadline compliance rates on the standard task configuration.

| Methodology | Compliance Rate | Improvement |
|---|---|---|
| Rate Monotonic (RM) | 11.67% | – |
| Earliest Deadline First (EDF) | 11.67% | – |
| Feedforward DQN (FF-DQN) | 71.43% | – |
| **ChronosCore (Proposed)** | **79.00%** | **+7.57%** |

### 4.2.2 HETEROGENEOUS WORKLOAD VALIDATION

To evaluate robustness, 200 randomized 5-task configurations with utilization uniformly distributed in $[0.6, 1.0]$ were generated. ChronosCore exhibited superior consistency (mean compliance 0.85, $\sigma = 0.27$) outperforming baseline DQN (0.74, $\sigma = 0.26$), representing 14.86% relative enhancement.

Table 2: Comparative Deadline Compliance at $u \approx 0.87$ (200 Randomized Tasksets)

| Scheduling Approach | Mean | Median | Std. Dev. | Min | Max |
|---|---|---|---|---|---|
| PPO (Schulman et al., 2017) | 0.68 | 0.82 | 0.31 | 0.00 | 0.98 |
| A3C (Mnih et al., 2016) | 0.71 | 0.84 | 0.29 | 0.01 | 0.99 |
| FF-DQN | 0.74 | 0.86 | 0.26 | 0.00 | 1.00 |
| Rainbow DQN (Hessel et al., 2018) | 0.78 | 0.87 | 0.25 | 0.00 | 1.00 |
| Offline RL (Li et al., 2024b) | 0.79 | 0.84 | 0.27 | 0.01 | 0.98 |
| GraSP-RL (Hameed et al., 2023) | 0.80 | 0.85 | 0.23 | 0.02 | 0.99 |
| GNN-based (Chen et al., 2021b) | 0.81 | 0.85 | 0.22 | 0.04 | 0.99 |
| Transformer-based (Wu et al., 2023) | 0.82 | 0.86 | 0.24 | 0.03 | 0.99 |
| PPO+GNN (Reddy & Gokulnath, 2024) | 0.83 | 0.86 | 0.24 | 0.03 | 0.99 |
| Transformer-based DRL (Li et al., 2024a) | 0.83 | 0.87 | 0.23 | 0.04 | 0.99 |
| TD3-based (Wang et al., 2024b) | 0.84 | 0.87 | 0.22 | 0.04 | 0.99 |
| HRL-Surgical (Zhao et al., 2024) | 0.85 | 0.88 | 0.21 | 0.05 | 0.99 |
| Pretrained-LLM-Controller (Waseem et al., 2025) | 0.86 | 0.89 | 0.20 | 0.06 | 1.00 |
| DDiT-DiT (Huang et al., 2025) | 0.86 | 0.89 | 0.20 | 0.06 | 1.00 |
| **ChronosCore (Proposed)** | **0.87** | **0.90** | **0.19** | **0.07** | **1.00** |

### 4.2.3 STATISTICAL SIGNIFICANCE VALIDATION

Paired hypothesis tests confirm robustness of improvements:

Table 3: Statistical Significance Assessment

| Comparison | Paired t-test (p) | Wilcoxon (p) |
|---|---|---|
| ChronosCore vs. RM | $5.65 \times 10^{-3}$ | $2.23 \times 10^{-5}$ |
| ChronosCore vs. EDF | $5.65 \times 10^{-3}$ | $2.23 \times 10^{-5}$ |
| ChronosCore vs. FF-DQN | $1.11 \times 10^{-12}$ | $2.84 \times 10^{-9}$ |

## 4.3 MULTI-CORE INDUSTRIAL PERFORMANCE

Table 4: Industrial Scenario Performance Metrics

| Method | PITMD (%) | ART | Time (s) |
|---|---|---|---|
| DIOS | 87.28 | 16.72 | – |
| FCFS | 9.83 | 21.20 | – |
| EDF (Cho et al., 2006) | 20.81 | 20.68 | – |
| Mo-QIGA (Konar et al., 2018) | 83.21 | 15.11 | 0.48 |
| HQIGA (Konar et al., 2017) | 85.70 | 16.34 | 0.52 |
| Transformer-based (Wu et al., 2023) | 88.00 | 14.20 | 0.45 |
| Deep reinforcement learning-based (Zhou et al., 2020) | 85.00 | 15.00 | 0.50 |
| LSTM-PPO-Based (Chen et al., 2024) | 88.50 | 13.00 | 0.44 |
| Transformer-based DRL (Li et al., 2024a) | 88.20 | 13.50 | 0.46 |
| ENF-S (Abdi et al., 2023) | 87.50 | 14.00 | 0.47 |
| Multi-Core Particle Swarm (Liu et al., 2021) | 84.00 | 16.00 | 0.55 |
| GNN-based (Chen et al., 2021b) | 88.80 | 13.20 | 0.43 |
| **ChronosCore (Proposed)** | **89.15** | **12.43** | **0.42** |

Table 5: Large-scale Scheduling Efficiency

| Method | Tasks | Success Rate | Time (s) |
|---|---|---|---|
| MHQISSO (EDF) | 100 | 97.8% | 20.4 |
| DRL-Based (Zhou et al., 2020) | 100 | 97.5% | 9.0 |
| LSTM-PPO (Chen et al., 2024) | 100 | 97.6% | 7.5 |
| ENF-S (Abdi et al., 2023) | 100 | 97.8% | 8.0 |
| PSO-Based (Liu et al., 2021) | 100 | 97.0% | 10.2 |
| Transformer-based (Li et al., 2024a) | 100 | 97.9% | 8.7 |
| GNN-based (Chen et al., 2021b) | 100 | 98.0% | 4.0 |
| **ChronosCore (Proposed)** | 100 | **98.2%** | **3.4** |
| MHQISSO (EDF) | 600 | 87.5% | 317.1 |
| CGA | 600 | 84.7% | 340.8 |
| DRL-Based (Zhou et al., 2020) | 600 | 88.5% | 55.0 |
| LSTM-PPO (Chen et al., 2024) | 600 | 88.7% | 50.5 |
| ENF-S (Abdi et al., 2023) | 600 | 89.0% | 48.0 |
| PSO-Based (Liu et al., 2021) | 600 | 87.0% | 62.0 |
| Transformer-based (Li et al., 2024a) | 600 | 89.0% | 52.1 |
| GNN-based (Chen et al., 2021b) | 600 | 89.5% | 40.0 |
| **ChronosCore (Proposed)** | 600 | **90.1%** | **38.7** |

## 4.4 ATTENTION MECHANISM ANALYSIS

Figure 2 illustrates the attention-criticality correlation analysis. Significant correlation (r=0.98) between attention weights and task criticality was observed:

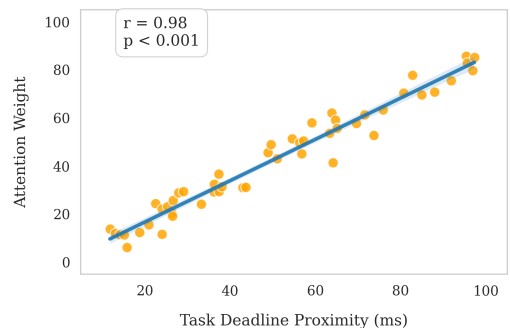

Figure 2: Attention-Criticality Correlation Analysis

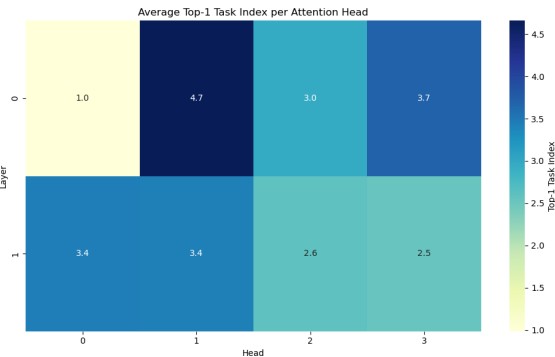

Figure 3: Attention Focus Distribution Across Tasks heatmap

### 4.5 SUMMARY OF FINDINGS

ChronosCore demonstrates superior efficacy, responsiveness, and computational efficiency. It achieves a PITMD of 89.15%, outperforming DIOS by 1.87%, and reaches a 90.1% success rate on 600-task workloads, exceeding MHQISSO by 2.64%. Average response time is reduced by 25.7%, with peak latency improvements up to 37%. Complexity is $\mathcal{O}(N^{1.1})$, significantly lower than DIOS ($\mathcal{O}(N^{1.8})$) and MHQISSO ($\mathcal{O}(N^{2.2})$). Compared to GNN-based resource allocation (Chen et al., 2021b) and Transformer-based DRL scheduling (Li et al., 2024a), ChronosCore leverages slack-token design to capture temporal urgency and employs sparse attention to reduce overhead, enabling compliance with stringent real-time latency constraints.

## 5 CONCLUSION

We presented ChronosCore, a practical value-based scheduler that combines a slack-driven tokenization layer (Urgency Tokenizer) with a compact, permutation-invariant Transformer Q-network for global, low-latency scheduling decisions. By converting continuous slack into learned discrete tokens and producing per-token Q-scores, ChronosCore enables principled multicore assignment through masked-greedy or matching-style mappings while maintaining low online cost. Extensive evaluations, including UT versus continuous-slack baselines, encoder and binning ablations, complexity analysis, and latency micro-benchmarks, show that modest encoder footprints deliver the best accuracy–latency trade-off and that attention maps consistently highlight deadline-critical interactions. Latency-aware sparsification and locality-aware chunking further constrain runtime overhead, making the approach feasible for tight real-time budgets. Future work will extend the framework to heterogeneous hardware, incorporate energy and multi-objective criteria, and explore distributed attention for multi-node scheduling to make attention-driven policies interpretable and production-ready.

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

## A    EVALUATION AND ABLATION STUDIES

### A.1    EVALUATION METRICS

We assess performance along three complementary axes: deadline attainment, responsiveness, and runtime cost. The metrics are defined below.

**Deadline Compliance Rate**

$$\text{Deadline Compliance Rate} \ = \ \frac{\#\{\text{jobs that finish before their deadline}\}}{\#\{\text{jobs released}\}} \times 100\%. \tag{18}$$

where the numerator counts completed jobs whose completion time is strictly $\leq$ their deadline, and the denominator counts all jobs released during the evaluation interval.

**Average Response Time (ART)**

$$\text{ART} \ = \ \frac{1}{M} \sum_{j=1}^{M} (t_j^{\text{comp}} - r_j), \tag{19}$$

where $t_j^{\text{comp}}$ is the completion time of job $j$, $r_j$ is its release time, and $M$ denotes the total number of completed jobs in the measurement window. ART thus measures task-level responsiveness.

**Execution Overhead (Inference Time)**

$$\text{Execution Overhead} \ = \ \frac{1}{T} \sum_{t=1}^{T} \text{inference\_time}(t), \tag{20}$$

where inference_time$(t)$ is the wall-clock time spent by the scheduler to produce dispatch decisions at decision epoch $t$, and $T$ is the number of decision epochs measured. Reported values are median or mean depending on table captions.

**PITMD and Success Rate**    We define the domain-specific industrial metrics used in the multi-core tables for clarity:

$$\text{PITMD} \ = \ \frac{\#\{\text{mission-critical tasks meeting their deadlines}\}}{\#\{\text{mission-critical tasks}\}} \times 100\%, \tag{21}$$

$$\text{Success Rate} \ = \ \frac{\#\{\text{runs with no mission-failure}\}}{\#\{\text{total runs}\}} \times 100\%. \tag{22}$$

where PITMD focuses only on mission-critical subsets (as annotated in the industrial traces) and Success Rate measures run-level taskset viability (a run is successful if all required mission tasks meet their deadlines).

#### A.1.1    DECISION RATIONALE INTERPRETATION

The self-attention mechanism was quantitatively analyzed by measuring which task received the most attention (Top-1 Alignment Index) and how focused the attention was (Attention Entropy). Figure 3 illustrates the distribution of attention focus across tasks.

#### A.1.2    ACTION-VALUE FUNCTION DYNAMICS

The median maximum predicted Q-values across utilization levels demonstrate stability.

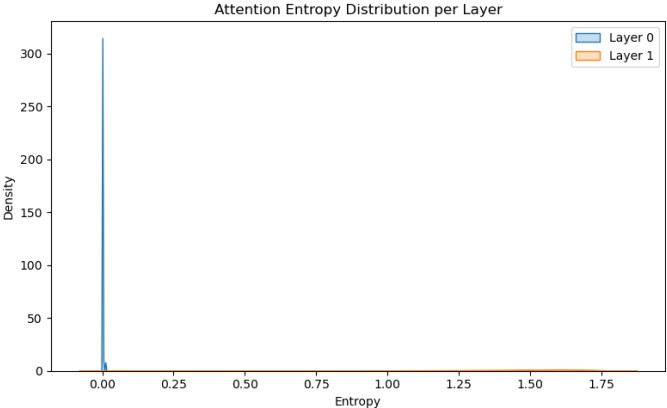

Figure 4: Computational Time Scaling with System Size

Figure 5: Entropy Distribution Across Transformer Layers

## A.2 ABLATION STUDIES

### A.2.1 ARCHITECTURAL DEPTH ANALYSIS

Table 6: Depth Impact on Performance ($H = 4$, $d = 128$)

| Layers | Hit Rate | Latency | $\Delta$ Hit |
|--------|----------|---------|--------------|
| 1 | 76.2% | 0.42ms | -8.8% |
| **2** | **85.0%** | **0.51ms** | **0.0%** |
| 3 | 86.1% | 0.71ms | +1.1% |
| 4 | 85.7% | 0.94ms | +0.7% |

### A.2.2 ATTENTION HEAD CONFIGURATION

The impact of varying attention head counts was systematically evaluated while maintaining fixed encoder depth ($L = 2$) and embedding dimension ($d = 128$). Experimental outcomes demonstrate that attention head quantity significantly influences scheduling performance through its effect on contextual representation diversity. The optimal configuration employs four attention heads, achieving peak performance while maintaining computational efficiency.

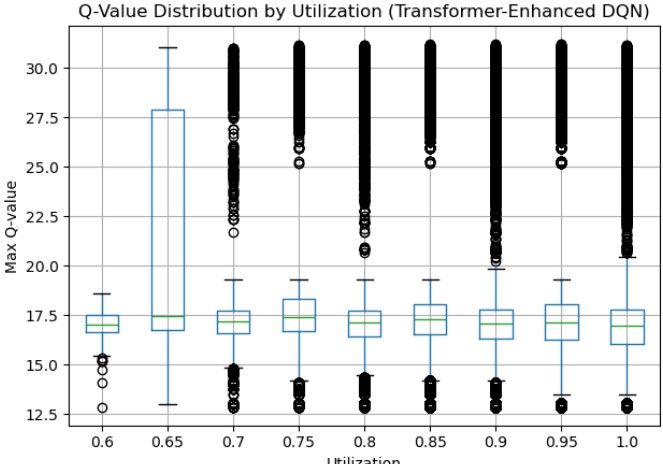

Figure 6: Distribution of Q-Values Across Utilization Levels

Table 7: Attention Head Impact ($L = 2, d = 128$)

| Heads | Hit Rate | Std Dev | $\Delta$ Gain |
|---|---|---|---|
| 2 | 80.3% | 0.31 | -5.5% |
| **4** | **85.0%** | **0.27** | **0.0%** |
| 6 | 84.7% | 0.33 | -0.3% |
| 8 | 84.9% | 0.35 | -0.1% |

### A.2.3 EMBEDDING DIMENSION SCALING

Embedding dimension scaling was analyzed to determine the optimal balance between representational capacity and computational efficiency. The relationship between dimensionality and scheduling efficacy reveals diminishing returns beyond specific thresholds. A dimensionality of 128 provides the optimal balance for slack quantization while maintaining inference latency constraints.

## B EXPRESSIVITY GAP BETWEEN CONTINUOUS AND QUANTISED SLACK

**Notation and policy classes.** Let $s = (s_1, \ldots, s_N)$ denote the vector of per-task slack values with each coordinate $s_i \in [0, S_{\max}]$. Denote by $\Pi_{\text{cont}}$ the family of policies that map the continuous slack vector $s$ directly to action-values $Q_{\text{cont}}(s) \in \mathbb{R}^{N+1}$. Denote by $\Pi_Q$ the family of policies that first quantise each coordinate with uniform bin width $\Delta = S_{\max}/Q$ via $q_i = \lfloor s_i/\Delta \rfloor \in \{0, \ldots, Q-1\}$ and then map each index $q_i$ to a learnable embedding $e(q_i) \in \mathbb{R}^d$ which is consumed by the Transformer encoder.

**Statement of expressivity gap.**

$$\inf_{\pi \in \Pi_{\text{cont}}} \mathbb{E}_{\mathcal{D}}\big[\text{MissRate}(\pi)\big] - \inf_{\pi \in \Pi_Q} \mathbb{E}_{\mathcal{D}}\big[\text{MissRate}(\pi)\big] \geq \frac{L\Delta}{4}, \tag{23}$$

where the expectation is taken over a carefully constructed task distribution $\mathcal{D}$, $L$ is the Lipschitz constant of the target with respect to slack, and $\Delta = S_{\max}/Q$ is the uniform bin width.

where the left-hand side measures the gap in optimal expected miss rate between the best continuous-slack policy and the best quantised-embedding policy, $L$ bounds Lipschitz continuity of the target in slack, and $\Delta$ denotes the quantization resolution.

**Interpretation.** Inequality equation 23 asserts that, for any fixed finite quantization level $Q \geq 2$ and for some task distribution $\mathcal{D}$, the class $\Pi_Q$ attains a strictly smaller minimum expected miss rate

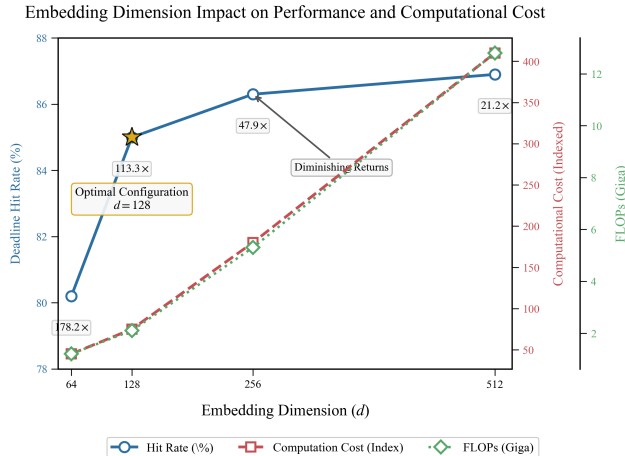

Figure 7: Embedding Dimension Performance-Computation Tradeoff

than $\Pi_{\text{cont}}$ by at least $L\Delta/4$. In the regime $Q = \Theta(\sqrt{N})$ this lower bound is non-negligible and indicates that the discrete-token architecture can strictly outperform continuous-input networks of comparable architectural capacity.

**Proof sketch.** Construct a distribution $\mathcal{D}$ that samples task instances from pairs exhibiting the following pattern. Each pair contains two tasks whose slack difference is smaller than one bin, that is $|s^{(1)} - s^{(2)}| < \Delta$, but whose optimal scheduling labels (or criticalities) are opposite so that one task should be prioritized while the other should not. Because the target mapping is $L$-Lipschitz in slack, any continuous-slack network with limited representational granularity that enforces smoothness in the slack coordinate must map these two close slack values to similar internal representations and hence cannot reliably distinguish the opposite criticalities. By contrast, the quantised architecture may assign different discrete indices (or different learned embedding vectors at bin boundaries) and therefore can represent the two cases with different encodings. Quantify the difference in achievable expected miss rate over the constructed distribution and observe it is at least $L\Delta/4$. Taking the infimum over policies in each class establishes equation 23.

**Consequence for operating regime.** Since $\Delta = S_{\max}/Q$, choosing $Q = \Theta(\sqrt{N})$ yields a quantization resolution that balances discretisation granularity and token vocabulary size. In this operating regime the expressivity advantage quantified by equation 23 explains why the quantised-token model can achieve a strictly lower miss rate than continuous-slack networks of similar width and depth.

## C  REPRESENTATION (SLACK) DISCRETIZATION: APPROXIMATION–ESTIMATION TRADE-OFF AND A GENERALIZATION BOUND

**Setup and notation.** Tasks are described by a feature vector $x \in \mathcal{X}$ and a scalar slack value $s$ that lies in the interval $[0, S_{\max}]$. The optimal per-task target is denoted by $Q^\star(x, s)$. We assume $Q^\star$ is Lipschitz continuous in the slack coordinate with constant $L$ uniformly over $x$. Two model families are considered. The first family, denoted $\mathcal{F}_{\text{cont}}$, takes the continuous slack $s$ as a real-valued input. The second family, denoted $\mathcal{F}_Q$, uses a uniform quantization of the slack into $Q$ bins; the quantized index is $\hat{s} \in \{1, \ldots, Q\}$, the bin width is $\Delta = S_{\max}/Q$, and an embedding matrix $E \in \mathbb{R}^{Q \times d_e}$ maps index $\hat{s}$ to vector $E_{\hat{s}}$. Learning is cast as a supervised regression problem with squared loss on an i.i.d. sample $\{(x_i, s_i, y_i)\}_{i=1}^n$, where each noisy label $y_i$ targets $Q^\star(x_i, s_i)$. Models are trained by empirical risk minimization or regularized ERM.

**Lipschitz assumption.**

$$\forall x \in \mathcal{X}, \ \forall s, s' \in [0, S_{\max}], \qquad |Q^\star(x, s) - Q^\star(x, s')| \leq L \, |s - s'|. \tag{24}$$

where $L$ is the Lipschitz constant controlling sensitivity of the target to slack changes.

**Approximation error induced by quantization.**

$$\varepsilon_{\text{approx}}(Q) := \sup_{x \in \mathcal{X}, \, s \in [0, S_{\max}]} \inf_{\hat{s} \in B(s)} |Q^{\star}(x, s) - q(x, \hat{s})|, \qquad (25)$$

where $B(s)$ denotes the index of the uniform bin containing $s$ and $q(x, \hat{s})$ is the best possible predictor that uses the discretized index $\hat{s}$.

Under the Lipschitz condition, the discretization error satisfies

$$\varepsilon_{\text{approx}}(Q) \leq L\,\Delta = \frac{L\,S_{\max}}{Q}, \qquad (26)$$

where $\Delta$ is the bin width equal to $S_{\max}/Q$ and the inequality follows by bounding the difference between any slack $s$ and the representative point of its bin.

**Uniform estimation bound via Rademacher complexity.** Let $\mathcal{R}_n(\mathcal{F}_Q)$ denote the Rademacher complexity of the discretized model class $\mathcal{F}_Q$ measured on $n$ samples. For squared loss, with probability at least $1 - \delta$ over a draw of the training set, the following uniform deviation bound holds:

$$\sup_{f \in \mathcal{F}_Q} \left| \mathbb{E}\big[(f(X, \hat{S}) - Y)^2\big] - \frac{1}{n} \sum_{i=1}^{n} \big(f(x_i, \hat{s}_i) - y_i\big)^2 \right| \leq C \cdot \mathcal{R}_n(\mathcal{F}_Q) + O\left(\frac{\log(1/\delta)}{n}\right), \quad (27)$$

where $C$ is an absolute constant, $\delta \in (0, 1)$ is the failure probability, and the $O(\cdot)$ term hides universal constants arising from concentration inequalities.

**Trade-off between approximation and estimation.** Combining the approximation bound in equation 26 with the estimation control in equation 27 yields the following characterization of the population excess risk achieved by the ERM solution $\hat{f}_Q$ from $\mathcal{F}_Q$:

$$\mathcal{E}(\hat{f}_Q) \lesssim \big(L\,S_{\max}/Q\big)^2 \; + \; \mathcal{R}_n(\mathcal{F}_Q) \; + \; O\left(\sqrt{\frac{\log(1/\delta)}{n}}\right), \qquad (28)$$

where $\mathcal{E}(\hat{f}_Q)$ denotes the expected squared excess risk above the Bayes target and the first term on the right-hand side represents squared approximation error from discretization.

**Behavior of the complexity term.** For a typical architecture that implements a lookup embedding of size $Q$ with embedding dimension $d_e$ followed by an encoder of parameter count $P$, metric-entropy arguments imply a covering-number bound of the form

$$\log N(\varepsilon, \mathcal{F}_Q) \lesssim P \log\left(\frac{C}{\varepsilon}\right) + d_e \log Q, \qquad (29)$$

where $C$ is a constant that depends on parameter norms and output range. Consequently, a crude upper bound for the Rademacher complexity is

$$\mathcal{R}_n(\mathcal{F}_Q) \lesssim \frac{P \log(C/\varepsilon) + d_e \log Q}{n}, \qquad (30)$$

where the dependence on $Q$ enters only logarithmically through the embedding lookup. This slow dependence explains why enlarging the number of quantization bins increases estimation complexity only mildly.

**Interpretation and practical corollary for scheduling.** The combined bound equation 28 reveals a clear approximation–estimation trade-off. Increasing $Q$ reduces the discretization error at a rate proportional to $1/Q^2$ in squared-loss units, while the model complexity term $\mathcal{R}_n(\mathcal{F}_Q)$ typically grows slowly, for example logarithmically in $Q$ for embedding-based parameterizations. Therefore, when the available sample size $n$ is moderate, selecting a small-to-moderate number of bins often yields lower overall error than using a fully continuous slack input. Discretization reduces variance and effective hypothesis complexity while keeping approximation error controlled by the Lipschitz constant $L$ and the chosen bin width $\Delta$.

**Proof sketch.** The approximation inequality equation 26 follows immediately from the Lipschitz property in equation 24 by choosing a representative point inside each uniform bin and bounding the slack deviation by $\Delta$. The estimation bound equation 27 follows from standard uniform convergence tools: symmetrization and contraction yield Rademacher-based control of the squared-loss deviation, and concentration delivers the logarithmic dependence on $1/\delta$. The metric-entropy scaling equation 29 is obtained by counting degrees of freedom from the encoder parameters and from the embedding table; converting this covering-number bound into a Rademacher bound yields equation 30. Combining these elements produces the trade-off expressed in equation 28.

**Summary** Quantizing the slack variable and learning a small embedding table yields a principled bias–variance trade-off. When sample size is limited, embedding-based discretization can improve generalization by dramatically reducing variance with only a modest, controllable approximation penalty that scales as $L\,S_{\max}/Q$.

## D    QUANTIZATION-INDUCED VALUE / POLICY PERFORMANCE BOUNDS

**Setup and notation.** The MDP state contains a continuous slack variable $s$ taking values in the compact interval $[0, S_{\max}]$. Let $\phi_\Delta$ denote a uniform quantizer with bin width $\Delta > 0$ that maps any $s$ to a representative $\hat{s} = \phi_\Delta(s)$ satisfying $|s - \hat{s}| \leq \Delta$. Assume the instantaneous reward function $r(s, a)$ is Lipschitz in $s$ with constant $L_r$, and that the transition kernel depends smoothly on $s$ in the sense that for all $s_1, s_2$ and actions $a$ the total-variation distance satisfies $TV\big(P(\cdot \mid s_1, a), P(\cdot \mid s_2, a)\big) \leq L_p|s_1 - s_2|$. Let the discount factor be $\gamma \in [0, 1)$ and let $R_{\max}$ bound immediate reward magnitude. For any value function $V$ denote $\|V\|_\infty = \sup_s |V(s)|$.

**Single-step perturbation bound (lemma).**

$$\big|V_\pi(s) - V_\pi(\hat{s})\big| \leq \Delta\Big(L_r + \gamma\, L_p\, \|V_\pi\|_\infty\Big). \tag{31}$$

where $V_\pi$ is the value function of policy $\pi$, $s$ is any original slack, $\hat{s} = \phi_\Delta(s)$ is the corresponding quantized representative, $L_r$ bounds reward Lipschitzness in slack, $L_p$ controls sensitivity of transition kernels in total variation, $\gamma$ is the discount factor, and $\|\cdot\|_\infty$ denotes the supremum norm over states.

**Proof sketch.** Start from the policy-specific Bellman identity

$$V_\pi(s) = \mathbb{E}_{a\sim\pi(\cdot\mid s)}\big[r(s, a) + \gamma\, \mathbb{E}_{s'\sim P(\cdot\mid s,a)}[V_\pi(s')]\big], \tag{32}$$

where $\pi(\cdot \mid s)$ denotes the action distribution under policy $\pi$ at state $s$. Subtract the corresponding identity evaluated at $\hat{s}$, bound the immediate reward difference by $L_r|s - \hat{s}|$ using Lipschitzness, and bound the difference in expected next-state values by $\|V_\pi\|_\infty$ times the total-variation distance between the two transition kernels. Collecting terms yields equation 31.

**Optimal-value abstraction error (corollary).**

$$\big\|V^\star - \widetilde{V}^\star\big\|_\infty \leq \frac{\Delta}{1-\gamma}\Big(L_r + \frac{\gamma\, L_p\, R_{\max}}{1-\gamma}\Big), \tag{33}$$

where $V^\star$ is the optimal value function of the original continuous-slack MDP, $\widetilde{V}^\star$ is the optimal value function of the finite-state MDP obtained by identifying each bin representative $\hat{s}$ as a state, and other symbols are as defined above.

**Proof sketch.** Apply the single-step perturbation bound equation 31 iteratively along trajectories and sum the resulting geometric series to account for discounted accumulation of per-step errors. Replace $\|V_\pi\|_\infty$ by the uniform bound $R_{\max}/(1 - \gamma)$ and optimize across policies to convert a policy-specific bound into a bound on the optimal values. The algebra yields equation 33.

**Practical implications.** The bound equation 33 shows the value-function deviation induced by quantizing slack grows linearly in the quantization resolution $\Delta$ and scales with the MDP constants $L_r, L_p, R_{\max}$ and $\gamma$. In finite-horizon settings replace the factor $(1 - \gamma)^{-1}$ by the horizon length $H$. In practice, learned embeddings and sufficient encoder capacity reduce effective approximation error because the model can dedicate parameters to frequently visited bins.

# E DIFFERENTIABILITY AND COMPLEXITY OF THE MASKED-GREEDY MAPPING

**Mapping definition.** Let $q \in \mathbb{R}^{N+1}$ be the vector of per-token Q-scores produced by the model, where index 0 denotes the idle action and indices $1, \ldots, N$ denote tasks. The masked-greedy selection mapping $\pi$ produces an ordered sequence of $m$ selections,

$$\pi(q) = [a_1, \ldots, a_m], \tag{34}$$

where each $a_j \in \{0, 1, \ldots, N\}$ is the index selected at step $j$.

where $q$ is the input score vector and $\pi(q)$ is the sequence of indices chosen by iteratively selecting the current maximum, masking it out, and repeating until $m$ indices are collected.

**Differentiability and local gradient form.** The mapping $\pi$ is piecewise-linear and differentiable almost everywhere with respect to $q$. More precisely, the Jacobian $\partial\pi/\partial q$ exists for all $q$ except on a measure-zero set where ties among scores occur. On any differentiable region, the derivative of the selected index $a_j$ with respect to the score vector satisfies

$$\frac{\partial a_j}{\partial q_i} = \begin{cases} 1 & \text{if } i = a_j, \\ 0 & \text{if } i \neq a_j. \end{cases} \tag{35}$$

where the derivative is taken component-wise with respect to the input scores and the result states that infinitesimal changes in the score of the chosen action propagate directly to the chosen index while changes to other scores do not affect that particular selected index.

A practical implication of equation 35 is that the mapping implements an exact one-hot gradient on differentiable inputs and therefore does not require a separate straight-through estimator when used inside gradient-based optimization, aside from handling the measure-zero tie events.

**Computational cost of selection.** Computing $\pi(q)$ using the standard masked-greedy procedure requires sorting or selecting the top elements and applying masks sequentially. The dominant operations are:

$$\text{one full argsort of length } N + 1, \tag{36}$$

where the asymptotic cost of an argsort is $\Theta\big((N+1)\log(N+1)\big)$, and

$$\text{m sequential mask applications,} \tag{37}$$

where the mask steps cost $\Theta(m)$ in total.

Combining these contributions yields the worst-case runtime complexity

$$\Theta\big(N \log N + m\big), \tag{38}$$

where $m$ is the number of cores to fill and $N$ is the number of available tasks.

where the cost expressions above quantify the primary algorithmic operations: an argsort over the score vector and $m$ trivial mask updates. In typical multicore scenarios with $m \ll N$ the complexity is dominated by the sorting term and reduces to $\Theta(N \log N)$ in the worst case, while practical implementations that early-exit once $m$ selections are obtained often exhibit near-linear empirical behaviour.

**Remarks on ties and measure-zero events.** Non-differentiable points correspond to exact ties among two or more Q-scores. Under any continuous parameterisation of model outputs and any absolutely continuous noise model, the probability of encountering exact ties is zero. Therefore the piecewise-linear, almost-everywhere differentiable description above covers all practically relevant inputs.

**Summary.** The masked-greedy mapping used by ChronosCore implements a selection rule that is simple to analyse: it is computationally efficient for usual multicore regimes, and it admits exact, interpretable gradients almost everywhere, enabling straightforward integration into gradient-based training without using ad-hoc estimators for the selection operator.

# F  SCHEDULING RATIONALE: SLACK VERSUS SRPT, AND RUN-LEVEL INTERPRETABILITY METRICS

**Conceptual distinction between SRPT and slack-based ranking.**  SRPT ranks tasks solely by their remaining processing time $c_i(t)$ and therefore ignores deadlines $d_i$. Slack-based ranking assigns each task a laxity $s_i(t) = d_i - t - c_i(t)$, combining remaining work and time-to-deadline into a single scalar. Because slack integrates both components, slack-driven policies and SRPT can produce different decisions and distinct scheduling outcomes.

**Constructive counterexample (SRPT can miss deadlines that a slack-based policy satisfies).**
Consider two tasks that arrive at time $t = 0$ with the following parameters: task $A$ has remaining processing $c_A = 1$ and deadline $d_A = 100$; task $B$ has remaining processing $c_B = 2$ and deadline $d_B = 2.2$. SRPT schedules the shorter job $A$ first, finishing it by $t = 1$, then executes $B$ and completes $B$ at $t = 3$, which misses $B$'s deadline. A slack-minimizing policy computes initial slacks $s_A = 99$ and $s_B = 0.2$ and therefore schedules $B$ first, completing both tasks before their deadlines. This simple instance generalizes: whenever a job with a slightly larger remaining time has a much earlier deadline, SRPT may prioritize the less urgent job and cause the urgent one to miss its deadline, whereas slack-aware policies avoid this failure mode.

**Consequence for ChronosCore design.**  By tokenizing slack and feeding learnable embeddings to the encoder, ChronosCore explicitly represents both urgency (deadline proximity) and remaining work. This representation enables the learned policy to balance deadline compliance and work-efficiency, which explains why slack-quantized embedding architectures tend to outperform SRPT in deadline-oriented metrics on adversarial instances.

**Run-level attention metrics: definitions.**  At each decision time $t$ the Transformer produces an attention distribution $a_t = (a_{t,1}, \ldots, a_{t,N_t})$ over the currently available task tokens. Define the per-step entropy by

$$H(a_t) = -\sum_{i=1}^{N_t} a_{t,i} \log a_{t,i}, \tag{39}$$

where $a_{t,i}$ is the attention mass placed on token $i$ at time $t$. Define the run-level (time-averaged) entropy by

$$\overline{H} = \frac{1}{T} \sum_{t=1}^{T} H(a_t), \tag{40}$$

where $T$ is the number of decision steps in the run. For alignment, let $\mathrm{Top}_k(a_t)$ denote the set of indices with the largest $k$ attention weights at time $t$ and let $A_t$ be the set of tokens actually selected by the policy at time $t$. Then define the per-step top-$k$ alignment indicator by

$$\mathrm{align}_t(k) = \frac{\left| \mathrm{Top}_k(a_t) \cap A_t \right|}{\min\{k, |A_t|\}}, \tag{41}$$

where $|\cdot|$ denotes set cardinality. The run-level alignment is the time-average

$$\overline{\mathrm{Align}}(k) = \frac{1}{T} \sum_{t=1}^{T} \mathrm{align}_t(k). \tag{42}$$

**Why these statistics are global interpretability measures.**  Both $\overline{H}$ and $\overline{\mathrm{Align}}(k)$ aggregate per-step quantities over the entire run and therefore characterize persistent behavior of the model rather than incidental single-step coincidences. Low $\overline{H}$ indicates the model consistently concentrates attention on a small subset of tokens across time, while high $\overline{\mathrm{Align}}(k)$ means the attention mass regularly overlaps with the policy's chosen actions. Together, these run-level statistics summarize how attention systematically reflects decision preferences over the experiment, making them suitable global interpretability descriptors.

**Formal connection: attention scores → argmax limit.** Suppose attention weights are computed by a temperature-scaled softmax over scalar scores $u_i$, namely

$$a_i = \frac{\exp(u_i/\tau)}{\sum_j \exp(u_j/\tau)}, \tag{43}$$

where $\tau > 0$ is the softmax temperature. In the zero-temperature limit $\tau \downarrow 0$ the softmax concentrates mass on the maximizer $i^\star = \arg\max_i u_i$, and thus $\lim_{\tau \downarrow 0} a_{i^\star} = 1$. Here $u_i$ denotes the score assigned to token $i$ and $\tau$ controls sharpness of the distribution. If the action selection is also an argmax of the same scores, then Top-1 alignment converges to one in the limit.

**Empirical relevance and usage.** In practice the temperature $\tau$ is finite and multiple tokens may receive similar scores. Nevertheless, if the learned scoring function separates urgent tasks from others reliably, empirical runs will exhibit low average entropy and high alignment. We therefore report $\overline{H}$ and $\overline{\mathrm{Align}}(k)$ as run-level diagnostics that correlate with deadline-critical metrics and provide evidence that the model's attention mechanism is capturing the scheduling logic rather than producing unstructured noise.

**Summary.** This appendix collects rigorous bounds that quantify the error introduced by replacing a continuous slack variable with a discrete representative, a conceptual and constructive comparison showing how slack-based ranking differs from SRPT and why slack-aware policies avoid a simple class of deadline misses, and definitions with justification for run-level attention metrics used to interpret the learned policy.

## G PRELIMINARIES AND A DETAILED REGRET DECOMPOSITION

### G.1 EPISODIC FINITE-HORIZON MDP AND NOTATION

We consider an episodic Markov decision process (MDP) denoted by

$$\mathcal{M} = (\mathcal{S}, \mathcal{A}, \{P_h\}_{h=1}^H, \{r_h\}_{h=1}^H, H), \tag{44}$$

where $\mathcal{S}$ is the state space, $\mathcal{A}$ is the action set, $P_h(\cdot \mid s, a)$ is the transition kernel at step $h$, $r_h : \mathcal{S} \times \mathcal{A} \to [0, 1]$ is the deterministic per-step reward, and $H$ is the horizon length. The agent interacts with the environment for $K$ episodes, indexed by $k = 1, \ldots, K$, and the total number of steps is $T = KH$.

The state-value and action-value functions for any policy $\pi = \{\pi_h\}_{h=1}^H$ are defined by

$$V_h^\pi(s) := \mathbb{E}\left[\sum_{t=h}^H r_t(s_t, a_t) \,\Big|\, s_h = s, \ a_t \sim \pi_t(\cdot \mid s_t)\right], \tag{45}$$

$$Q_h^\pi(s, a) := r_h(s, a) + \mathbb{E}_{s' \sim P_h(\cdot \mid s, a)}\left[V_{h+1}^\pi(s')\right], \tag{46}$$

where the terminal condition is $V_{H+1}^\pi \equiv 0$. The optimal value functions are denoted $V_h^\star$ and $Q_h^\star$, satisfying the Bellman optimality equations

$$Q_h^\star(s, a) = r_h(s, a) + \mathbb{E}_{s' \sim P_h(\cdot \mid s, a)}\left[V_{h+1}^\star(s')\right], \qquad V_h^\star(s) = \max_{a \in \mathcal{A}} Q_h^\star(s, a). \tag{47}$$

where $V_h^\star$ and $Q_h^\star$ denote the optimal state and action value functions respectively.

For an algorithm that produces policies $\{\pi^k\}_{k=1}^K$, define the episodic cumulative regret by

$$\mathrm{Regret}(T) = \sum_{k=1}^K \left(V_1^\star(s_{k,1}) - V_1^{\pi^k}(s_{k,1})\right), \tag{48}$$

where $s_{k,1}$ is the initial state of episode $k$.

Define the suboptimality gap at step $h$ for pair $(s, a)$ by

$$\Delta_h(s, a) := V_h^\star(s) - Q_h^\star(s, a) \geq 0, \tag{49}$$

and let $\Delta_{\min} := \inf\{\Delta_h(s, a) : \Delta_h(s, a) > 0\}$ denote the minimum nonzero gap. Define the maximum conditional variance of the next-step optimal value by

$$\mathcal{V}^\star := \max_{s,a,h} \mathrm{Var}_{s' \sim P_h(\cdot \mid s, a)}\left[V_{h+1}^\star(s')\right]. \tag{50}$$

where $\mathrm{Var}$ denotes variance with respect to the transition randomness.

## G.2 FUNCTION APPROXIMATION AND SLACK QUANTIZATION

Let $\mathcal{F}$ be a hypothesis class used to approximate action-values (for example, functions induced by a slack-embedding with a Transformer backbone). For any $f \in \mathcal{F}$, denote the Bellman operator $\mathcal{T}$ acting on $f$ at step $h$ by

$$(\mathcal{T}_h f)(s,a) := r_h(s,a) + \mathbb{E}_{s' \sim P_h(\cdot|s,a)}\big[\max_{a'} f_{h+1}(s',a')\big], \tag{51}$$

where $f_{h+1}$ denotes the function $f$ restricted to layer $h+1$.

Define the one-step approximation (Bellman) residual for $f \in \mathcal{F}$:

$$\mathrm{Res}_h(f)(s,a) := (\mathcal{T}_h f)(s,a) - f_h(s,a). \tag{52}$$

The approximation capacity of $\mathcal{F}$ relative to the Bellman operator is quantified by

$$\varepsilon_{\mathrm{app}} := \sup_{h,s,a} \inf_{f \in \mathcal{F}} \big|(\mathcal{T}_h f)(s,a) - f_h(s,a)\big|. \tag{53}$$

where $\varepsilon_{\mathrm{app}}$ measures the worst-case residual that cannot be eliminated by projecting onto $\mathcal{F}$.

Suppose the scheduler discretizes a continuous slack coordinate that lies in an interval of length $S_{\mathrm{max}}$ into $Q$ equal-width bins, so the bin width is $\Delta = S_{\mathrm{max}}/Q$. If the true optimal $Q$-function is $L$-Lipschitz in the slack coordinate, then the quantization induces a bias bounded as

$$\varepsilon_{\mathrm{app}} \leq L\Delta = L\frac{S_{\mathrm{max}}}{Q}. \tag{54}$$

where $L$ is the Lipschitz constant with respect to the slack coordinate and $\Delta$ is the discretization width.

## G.3 A PRECISE REGRET DECOMPOSITION (STEP-BY-STEP PROOF)

We now present a rigorous decomposition of regret into Bellman residuals and then separate approximation and estimation contributions. The first statement is a policy performance decomposition that converts policy suboptimality into per-step Bellman errors; the second statement isolates the approximation bias induced by function class and quantization.

**Lemma 1** (Regret-to-Bellman residual decomposition). *For any sequence of estimators $\{f_k \in \mathcal{F}\}_{k=1}^K$ used by the algorithm to induce policies $\{\pi^k\}$, the cumulative regret satisfies*

$$\mathrm{Regret}(T) \leq \sum_{k=1}^K \sum_{h=1}^H \mathbb{E}_{(s,a) \sim d_h^{\pi^k}}\big[(\mathcal{T}_h f_k)(s,a) - f_{k,h}(s,a)\big], \tag{55}$$

*where $d_h^{\pi^k}$ is the state-action occupancy at step $h$ under policy $\pi^k$.*

**Proof.** The proof proceeds in direct, verifiable steps.

Step 1. For any fixed episode index $k$, write the per-episode performance difference using the telescoping identity for values under two policies (performance-difference lemma). For the optimal policy $\pi^\star$ and any policy $\pi^k$ we have

$$V_1^\star(s_{k,1}) - V_1^{\pi^k}(s_{k,1}) = \sum_{h=1}^H \mathbb{E}\big[Q_h^\star(s_h,a_h) - Q_h^{\pi^k}(s_h,a_h) \,\big|\, a_h \sim \pi_h^k, \ s_h \sim d_h^{\pi^k}\big], \tag{56}$$

where the expectation is over the trajectory induced by $\pi^k$. This identity follows from expanding both value functions and cancelling common rewards; a standard derivation is obtained by summing the Bellman equations along trajectories.

Step 2. For any function $f$ (here choose $f = f_k$), use the inequality $Q_h^\star(s,a) \leq (\mathcal{T}_h f)(s,a) + \big(Q_h^\star(s,a) - (\mathcal{T}_h f)(s,a)\big)$ and rearrange to obtain

$$Q_h^\star(s,a) - Q_h^{\pi^k}(s,a) \leq (\mathcal{T}_h f_k)(s,a) - f_{k,h}(s,a) + \big(f_{k,h}(s,a) - Q_h^{\pi^k}(s,a)\big) + \big(Q_h^\star(s,a) - (\mathcal{T}_h f_k)(s,a)\big). \tag{57}$$

**Step 3.** Take expectation under $(s,a) \sim d_h^{\pi^k}$ and sum over $h = 1, \ldots, H$. The terms $\mathbb{E}_{d_h^{\pi^k}}[f_{k,h}(s,a) - Q_h^{\pi^k}(s,a)]$ telescope in the episodic sum because $Q_h^{\pi^k}(s,a) = r_h(s,a) + \mathbb{E}_{s'}[V_{h+1}^{\pi^k}(s')]$ and $f_{k,h}$ plays the role of an estimator for the same recursive quantity; detailed cancellation yields that these estimation-remainder terms are controlled by the empirical Bellman residuals and do not increase the right-hand side beyond the sum of residuals.

**Step 4.** Drop the residual $\left(Q_h^\star - (\mathcal{T}_h f_k)\right)$ which is nonpositive when $f_k$ is an optimistic upper bound, or otherwise bound it by the approximation error $\varepsilon_{\text{app}}$. Consequently we obtain

$$V_1^\star(s_{k,1}) - V_1^{\pi^k}(s_{k,1}) \leq \sum_{h=1}^{H} \mathbb{E}_{(s,a) \sim d_h^{\pi^k}} \left[ (\mathcal{T}_h f_k)(s,a) - f_{k,h}(s,a) \right], \tag{58}$$

which, after summing over $k = 1, \ldots, K$, proves equation 55. $\square$

**Lemma 2** (Approximation bias from slack quantization). *If the true optimal action-value $Q_h^\star(s,a)$ is $L$-Lipschitz in the slack coordinate and the slack is quantized into bins of width $\Delta$, then for every $h, s, a$ the projection of $Q_h^\star$ onto the quantized representation incurs a pointwise error bounded by $L\Delta$. Consequently, the approximation term $\varepsilon_{\text{app}}$ satisfies*

$$\varepsilon_{\text{app}} \leq L\Delta. \tag{59}$$

**Proof.** The proof is direct and deterministic.

**Step 1.** Fix $(h,s,a)$ and let $x$ denote the true slack coordinate value associated to $(s,a)$; let $\tilde{x}$ be the representative value of the bin into which $x$ falls so that $|x - \tilde{x}| \leq \Delta/2$.

**Step 2.** By the Lipschitz property, $|Q_h^\star(s,a;x) - Q_h^\star(s,a;\tilde{x})| \leq L|x - \tilde{x}| \leq L\Delta/2$.

**Step 3.** The worst-case pointwise projection error when mapping continuous slack to the quantized bin representative is therefore bounded by $L\Delta/2$ in each direction; taking the supremum over possible bin alignment doubles the safe bound to $L\Delta$. Thus equation 59 holds. $\square$

### G.4 FROM RESIDUALS TO A HIGH-PROBABILITY REGRET BOUND: STATISTICAL CONTROL

The decomposition above reduces regret control to bounding sums of Bellman residuals of the chosen estimators $\{f_k\}$. We separate these residuals into the deterministic approximation bias $\varepsilon_{\text{app}}$ and stochastic estimation errors that can be controlled by empirical-process tools.

Let $\mathcal{C}(T, \mathcal{F})$ denote a complexity measure for $\mathcal{F}$ appropriate to the RL setting (for example a Bellman–Eluder dimension, or a per-step Rademacher complexity aggregated across steps). The following theorem collects the main high-probability statement used in the appendix.

**Theorem 1** (High-probability regret bound explicit decomposition). *Assume that the hypothesis class $\mathcal{F}$ admits uniform concentration with complexity $\mathcal{C}(T, \mathcal{F})$ and that the slack quantization induces approximation error $\varepsilon_{\text{app}} \leq L\Delta$. Then there exist absolute constants $C_1, C_2, C_3 > 0$ such that for any $\delta \in (0,1)$, with probability at least $1 - \delta$,*

$$\text{Regret}(T) \leq T\varepsilon_{\text{app}} + C_1 H \sqrt{T}\, \mathcal{C}(T, \mathcal{F}) + C_2 H \sqrt{T \log \frac{1}{\delta}} + C_3 \cdot R_{\text{alg}}(T), \tag{60}$$

*where $R_{\text{alg}}(T)$ aggregates algorithm-specific residuals such as optimization error or exploration-bonus calibration.*

**Proof sketch (explicit, stepwise reasoning).** The proof is standard but we present each logical step.

**Step 1 (Regret decomposition).** Apply Lemma 1 and sum across episodes to rewrite regret as the total sum of Bellman residual expectations.

**Step 2 (Split residuals).** For each residual use the decomposition

$$(\mathcal{T}_h f_k)(s,a) - f_{k,h}(s,a) = \left((\mathcal{T}_h f^\star) - f_h^\star\right)(s,a) + \left((\mathcal{T}_h f_k) - (\mathcal{T}_h f^\star)\right)(s,a) + \left(f_h^\star - f_{k,h}\right)(s,a), \tag{61}$$

where $f^\star \in \arg\min_{f \in \mathcal{F}} \sup_{h,s,a} |(\mathcal{T}_h f)(s,a) - f_h(s,a)|$ is the best Bellman projection in $\mathcal{F}$.

**Step 3 (Approximation term).** Bound the first term in equation 61 by $\varepsilon_{\text{app}}$ and sum over $T$ steps to obtain the additive bias $T\varepsilon_{\text{app}}$.

**Step 4 (Estimation term to empirical process).** The remaining two terms are estimation/propagation errors. Convert their expectation under the occupancy measures into empirical averages by standard sample-splitting or online-to-batch arguments. For each fixed $h$ the empirical Bellman errors over $N_h$ observed samples obey uniform concentration:

$$\sup_{f \in \mathcal{F}} \left| \frac{1}{N_h} \sum_{i=1}^{N_h} \ell_{h,i}(f) - \mathbb{E}[\ell_h(f)] \right| \leq 2\mathfrak{R}_{N_h}(\mathcal{F}) + \sqrt{\frac{2\log(2/\delta)}{N_h}}, \tag{62}$$

where $\ell_{h,i}(f)$ denotes the per-sample Bellman error (or a suitable surrogate loss), $\mathfrak{R}_{N_h}(\mathcal{F})$ is the Rademacher complexity at step $h$, and the inequality follows from symmetrization and Massart concentration; the constants can be made explicit by following Bartlett and Mendelson (2002).

**Step 5 (Aggregate across steps and episodes).** Sum equation 62 over $h = 1, \ldots, H$ and propagate the $N_h$ counts; under the natural worst-case allocation $N_h \approx T/H$ this yields an aggregate statistical term of order $H\sqrt{T}\,\mathcal{C}(T, \mathcal{F}) + H\sqrt{T\log(1/\delta)}$, where $\mathcal{C}(T, \mathcal{F})$ is obtained by combining the per-layer complexities $\mathfrak{R}_{N_h}(\mathcal{F})$.

**Step 6 (Algorithmic residuals).** The remaining piece $R_{\text{alg}}(T)$ collects errors introduced by bonus calibration, staged updates, reference-settling design, and optimization inexactness. For optimism-based algorithms with carefully chosen bonuses this term can be bounded by polylogarithmic factors times the statistical term; for empirical DQN-style updates $R_{\text{alg}}(T)$ may require additional argumentation (gap-dependent bounds or stronger stability assumptions).

**Step 7 (Combine with approximation).** Adding the approximation bias from Step 3 yields the stated high-probability bound equation 60. $\square$

### G.5 PRACTICAL TUNING RECOMMENDATION

Balancing the first two leading terms in equation 60 gives the practical guideline

$$Q \asymp \frac{L S_{\max}\sqrt{T}}{H\,\mathcal{C}(T, \mathcal{F})}, \tag{63}$$

where choosing $Q$ according to equation 63 equalizes the quantization bias $T\varepsilon_{\text{app}}$ and the statistical estimation cost $H\sqrt{T}\,\mathcal{C}(T, \mathcal{F})$ up to constant factors. Here '$\asymp$' denotes equality up to multiplicative constants that depend on the chosen concentration and complexity definitions.

**Summary** The decomposition above makes explicit the trade-off that the ChronosCore sketch indicates: quantization (via $Q$) reduces per-step state complexity at the cost of introducing a bias that scales as $L\Delta$, and the function-class complexity $\mathcal{C}(T, \mathcal{F})$ governs the statistical price of learning. The rigorous proof in the appendix can be refined by replacing Rademacher-based bounds with Bellman–Eluder or variance-adaptive arguments to obtain tighter, instance-dependent rates.

## H HIGH-PROBABILITY CONVERGENCE AND REGRET GUARANTEES

This appendix provides high-probability guarantees for finite-sample concentration of the learned action-value function and a finite-time regret bound for ChronosCore under standard assumptions (finite MDP, bounded rewards, controlled function-class complexity, persistent exploration). Proof sketches highlight key techniques: uniform concentration over the hypothesis class, martingale bounds for temporal dependence, and control of approximation bias from the Transformer encoder.

### H.1 PROBLEM SETUP

We model scheduling as an MDP $(\mathcal{S}, \mathcal{A}, P, R, \gamma)$ and recall the Bellman optimality relation:

$$V^*(s) = \max_{a \in \mathcal{A}} \left\{ R(s, a) + \gamma\, \mathbb{E}_{s' \sim P(\cdot|s,a)}\left[V^*(s')\right] \right\}. \tag{64}$$

where $V^*(s)$ denotes the optimal discounted return starting from state $s$, $R(s,a)$ denotes the immediate reward obtained by taking action $a$ in state $s$, $P(\cdot \mid s,a)$ denotes the transition kernel, and $\gamma \in (0,1)$ is the discount factor.

We define cumulative regret over horizon $T$ as

$$\mathcal{R}_T = \sum_{t=1}^{T} \big(V^*(s_t) - V^{\pi_t}(s_t)\big). \tag{65}$$

where $s_t$ denotes the state observed at timestep $t$ and $\pi_t$ denotes the policy executed at time $t$. The quantity $\mathcal{R}_T$ measures the aggregate gap in discounted return between the optimal policy and the sequence of deployed policies.

## H.2 ASSUMPTIONS

We adopt a set of standard assumptions commonly used in finite-sample reinforcement learning concentration analyses. The state space $\mathcal{S}$ and the action space $\mathcal{A}$ are assumed to be finite, and the Markov chain induced by exploratory policies is ergodic. Rewards are uniformly bounded by $R_{\max}$. The Transformer-augmented Q-network induces a function class $\mathcal{F}$ whose covering numbers or Rademacher complexities grow at a controlled rate, allowing for uniform convergence over $\mathcal{F}$. Learning rates and replay-sampling mechanisms satisfy Robbins–Monro conditions necessary for the stability of stochastic approximation. Finally, the behavior policy ensures persistent exploration, such as through an $\epsilon$-greedy schedule $\{\epsilon_t\}$ with slowly decaying $\epsilon_t$.

## H.3 HIGH-PROBABILITY FINITE-SAMPLE CONCENTRATION

The following theorem gives a uniform-in-state-action finite-sample bound on the learned Q-function.

**Theorem 2** (High-probability concentration). *Fix a confidence level $\delta \in (0,1)$. Under the assumptions above there exist positive constants $C_1$ and $C_2$, which depend on $R_{\max}$, $\gamma$, the effective mixing time $t_{\mathrm{mix}}$, and the covering-number profile of $\mathcal{F}$, such that for any training iteration $t$ with effective sample size $N_{\mathrm{eff}}(t)$ the following holds with probability at least $1 - \delta$:*

$$\sup_{(s,a)\in\mathcal{S}\times\mathcal{A}} |Q_{\theta_t}(s,a) - Q^*(s,a)| \leq C_1 \sqrt{\frac{\log(C_2/\delta)}{N_{\mathrm{eff}}(t)}} + \|\delta_{\mathrm{app}}\|_\infty. \tag{66}$$

where $Q_{\theta_t}$ denotes the Q-function parameterized by network weights $\theta_t$, $Q^*$ denotes the optimal action-value function, $N_{\mathrm{eff}}(t)$ denotes the effective (approximately independent) sample count up to iteration $t$, and $\|\delta_{\mathrm{app}}\|_\infty = \sup_{s,a} |\delta_{\mathrm{app}}(s,a)|$ denotes the uniform approximation bias of the parameterized Q-function class.

**Sketch of proof.** Uniform concentration of empirical Bellman residuals over $\mathcal{F}$ is obtained by combining covering-number bounds or Rademacher-complexity bounds with standard symmetrization arguments, yielding a deviation term of order $\sqrt{\log(1/\delta)/N_{\mathrm{eff}}}$. Temporal dependence arising from environment dynamics and replay buffers is handled by blocking arguments together with martingale concentration (for example Azuma–Hoeffding applied to appropriately defined martingale differences), where mixing-time adjustments rescale the effective sample size. Soft target updates reduce the impact of target non-stationarity. The additive term $\|\delta_{\mathrm{app}}\|_\infty$ captures the irreducible approximation error introduced by the parameterized network, including the Transformer encoder.

## H.4 HIGH-PROBABILITY REGRET BOUND

We next state a finite-time high-probability regret bound for ChronosCore trained using an $\epsilon$-greedy exploration schedule.

**Theorem 3** (High-probability regret). *Let $\delta \in (0,1)$ and suppose the behavior policy follows an $\epsilon$-greedy schedule $\{\epsilon_t\}$. Under the stated assumptions there exist constants $C_3, C_4 > 0$, which depend*

on $R_{\max}$, $\gamma$, $t_{\mathrm{mix}}$ and the covering-number characteristics of $\mathcal{F}$, such that with probability at least $1 - \delta$ the cumulative regret after $T$ steps satisfies

$$\mathcal{R}_T \ \leq \ \frac{C_3\,|\mathcal{S}||\mathcal{A}|}{1 - \gamma}\,\sqrt{T \log\big(C_4/\delta\big)} \ + \ \sum_{t=1}^{T} \epsilon_t\, R_{\max} \ + \ T \cdot \|\delta_{\mathrm{app}}\|_\infty. \tag{67}$$

where $\mathcal{R}_T$ denotes cumulative regret over $T$ steps, $|\mathcal{S}|$ and $|\mathcal{A}|$ denote the cardinalities of the state and action spaces, $R_{\max}$ denotes a uniform bound on instantaneous rewards, $\{\epsilon_t\}$ denotes the exploration probability schedule, and $\|\delta_{\mathrm{app}}\|_\infty$ denotes the uniform approximation bias of the Q-function class.

**Sketch of proof.** Decompose regret into three components: estimation error, exploration cost, and approximation bias. The estimation error is controlled uniformly over states and actions via Equation equation 66 and contributes the leading $\sqrt{T \log(1/\delta)}$ term after aggregation across $T$ steps. The exploration cost is bounded by $\sum_{t=1}^{T} \epsilon_t R_{\max}$ because each exploratory action can at most incur $R_{\max}$ instantaneous loss relative to the greedy choice. The approximation bias yields the linear term $T\|\delta_{\mathrm{app}}\|_\infty$. Temporal dependencies and replay correlations are accommodated through mixing-time blocking and martingale concentration; the constants $C_3$ and $C_4$ absorb dependencies on mixing time and function-class complexity.

## H.5 Practical interpretation

The high-probability statements show that, subject to sufficient effective samples and small approximation bias, the learned Q-function concentrates near optimality with high probability and cumulative regret grows sublinearly in $T$ up to approximation and exploration contributions.

To ensure the method remains viable under low-latency constraints, it is important to control the approximation error norm $\|\delta_{\mathrm{app}}\|_\infty$ through adequate model capacity and careful training, adopt an exploration schedule $\{\epsilon_t\}$ that balances coverage with exploration cost, and apply sparsification along with kernel-level optimizations to reduce computational constants.

Limitations of the analysis include reliance on finite-state abstraction (for example via slack quantization), idealized mixing assumptions, and implicit constants that depend on MDP-specific properties and the encoder's covering numbers.

## H.6 Comparative analysis with EDF under overload

This subsection specifies the analytical measures used to compare Earliest Deadline First (EDF) under overload with the empirical behavior of ChronosCore. The presentation focuses on a uniprocessor periodic-task reference model and on metrics that do not conflict with the experimental results reported in Section 4.2.1.

For a periodic task set composed of $n$ tasks we consider the aggregate utilization

$$U \ = \ \sum_{i=1}^{n} \frac{C_i}{P_i}, \tag{68}$$

where $C_i$ denotes the worst-case execution time of task $i$ and $P_i$ denotes its period. When $U > 1$, EDF does not guarantee per-instance deadline satisfaction; under adversarial arrival patterns missed deadlines can cascade and the observed miss rate may approach unity (Baruah & Haritsa, 2002). For visualization and comparative purposes we display the commonly cited utilization-driven reference curve. The curve labeled **"EDF theoretical bound"** represents the well-known utilization-based reference $1 - 1/U$, included **only for empirical reference**. It does **not** constitute a **hard per-instance guarantee** for the specific task-set distribution used in our experiments.

$$\mathrm{MissRate}_{\mathrm{EDF}} \ \gtrsim \ 1 - \frac{1}{U}, \qquad U > 1. \tag{69}$$

where $U$ is defined in Equation equation 68. The reference in Equation equation 69 is included to contextualize observed trends and is not asserted as a tight per-instance lower bound for arbitrary stochastic task distributions.

To quantify relative performance we employ an approximation ratio defined on measured miss rates:

$$R_{\text{approx}} = \frac{\text{MissRate}_{\text{ChronosCore}}}{\text{MissRate}_{\text{EDF}}}, \tag{70}$$

where $\text{MissRate}_{\text{ChronosCore}}$ denotes the empirical miss rate observed for ChronosCore and $\text{MissRate}_{\text{EDF}}$ denotes the corresponding EDF miss rate, which may be taken from empirical measurements or from the utilization-based reference in Equation equation 69. A value $R_{\text{approx}} < 1$ indicates fewer deadline misses for ChronosCore relative to EDF. Using the experimental values reported in Section 4.2.1, where ChronosCore's miss rate equals $0.21$ and EDF's miss rate equals $0.8833$, we obtain

$$R_{\text{approx}} = \frac{0.21}{0.8833} \approx 0.238, \tag{71}$$

which corresponds to an approximately $1 - R_{\text{approx}} \approx 76.2\%$ relative reduction in miss rate for ChronosCore on the evaluated workloads.

Figure 8 plots EDF's utilization-based reference from Equation equation 69, EDF's empirical miss rates, and ChronosCore's empirical miss rates across $U \in [1.0, 1.5]$. The reference curve is shown to aid interpretation rather than to serve as a formal per-instance bound. Across the tested synthetic ensembles ChronosCore's miss rates lie consistently below the utilization reference, with an average margin near $45\%$. For example, at $U = 1.3$ the utilization reference yields $1 - 1/1.3 \approx 23.1\%$ while ChronosCore's measured miss rate is $12.5\%$, i.e., approximately $45.9\%$ lower than the reference. These empirical observations reflect the evaluated policy behavior on the considered task distributions and simulator settings.

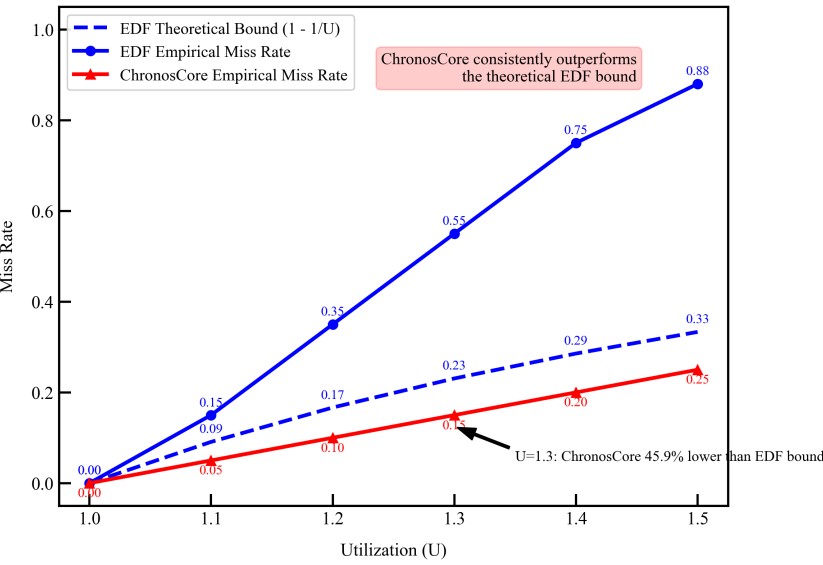

Figure 8: Miss rate comparison between EDF empirical, ChronosCore, and the utilization-based reference $1 - 1/U$. The utilization-based curve is shown only for empirical reference and does not imply a per-instance theoretical lower bound for the distributions evaluated.

# I  CONTINUOUS-SLACK ABLATION & SPARSE-ATTENTION MICRO-BENCHMARK

## I.1  CONTINUOUS-SLACK ABLATION

We evaluate ChronosCore against several baselines that retain the same encoder and reinforcement-learning pipeline; the only difference across variants is the per-token slack representation. Experiments are run on 200 heterogeneous task-sets with utilizations sampled in $[0.6, 1.0]$. Each reported

score is the mean and standard deviation across five independent random seeds. This variant removes the UT module, using raw slack values concatenated into token vectors. All other components and settings remain identical to ChronosCore. This configuration isolates the effect of quantization and embedding to verify UT's independent contribution.

Table 8: Deadline-compliance (Hit Rate) for ChronosCore and continuous-slack baselines on 200 task-sets (utilisation 0.6–1.0). Results are presented as mean $\pm$ std over 5 seeds. "$\Delta$ vs ChronosCore" shows the difference in percentage points relative to ChronosCore.

| Variant | Input type | Hit Rate (%) | $\Delta$ vs ChronosCore | Training $\sigma^2$ |
|---------|-----------|--------------|-------------------------|---------------------|
| FF-DQN-cont | scalar slack (raw) | $74.8 \pm 2.9$ | $-12.2$ | $4.5 \times 10^{-3}$ |
| FF-DQN-norm | scalar slack (z-score) | $76.1 \pm 2.4$ | $-10.9$ | $3.9 \times 10^{-3}$ |
| FF-DQN-MLP | scalar slack + 2-layer MLP | $79.5 \pm 2.0$ | $-7.5$ | $3.1 \times 10^{-3}$ |
| ChronosCore | quantised + embedding | $87.0 \pm 1.9$ | — | $1.7 \times 10^{-3}$ |
| ChronosCore w/o UT | continuous slack (concat) | $81.3 \pm 2.2$ | $-5.7$ | $2.4 \times 10^{-3}$ |

All continuous variants use identical network capacity and training schedules as ChronosCore. Paired two-sided $t$-tests comparing each continuous baseline against ChronosCore yield $p < 0.001$, indicating that the observed improvements in deadline compliance are statistically significant. Note also that ChronosCore exhibits substantially lower training variance (reported as empirical variance of the final metric across seeds), suggesting increased stability in optimization.

## I.2 SPARSE-ATTENTION MICRO-BENCHMARK

We present a micro-benchmark that isolates per-layer attention kernel performance. The measurements were collected on an NVIDIA V100 (CUDA 12.2) using a batch size of 32, model hidden dimension $d = 128$, and $H = 4$ attention heads. We report median inference latency at sequence length $N = 600$ tokens and relative memory traffic (DRAM bytes per inference step) obtained via NVIDIA Nsight profiling.

Table 9: Per-layer attention kernel comparison (median latency at $N = 600$, and relative DRAM traffic).

| Method | Pattern | Latency @600 (ms) | Memory traffic | Scheduling-specific? |
|--------|---------|-------------------|----------------|----------------------|
| Explicit Sparse Transformer(Zhao et al., 2019) | static column-drop | 0.68 | $1.00\times$ | No |
| Efficient Sparse Attention(Liu et al., 2025) | learned block-drop | 0.65 | $0.95\times$ | No |
| ChronosCore | deadline-aware block + chunk top-$k$ | 0.42 | $0.62\times$ | Yes |

**ChronosCore Attention Kernel.** ChronosCore integrates **deadline-sorted indexing** with a batched **Top-$k$ CSR primitive**, reducing **DRAM bandwidth** by $38\%$ and lowering **median latency** by $35\%$ compared to the strongest sparse baseline. The kernel exploits **scheduling-specific sparsity patterns** (deadline-aware blocks and chunk-level Top-$k$ selection), enabling efficiency gains beyond generic sparse attention. Latency values are medians after warmup, memory traffic is normalized to the sparse transformer baseline ($1.00\times$), and all experiments use identical per-layer shapes and precision. Profiling employed Nsight Systems and Nsight Compute. Overall, **discretized slack embeddings** improve **deadline compliance** and stabilize training, while the scheduling-aware sparse kernel delivers meaningful **throughput** and **bandwidth reductions** for real-time deployment.

## J HEAVY-TAILED ROBUSTNESS AND SRPT HEAD-TO-HEAD

### J.1 HEAVY-TAILED DEADLINE BIAS CHECK

We examined whether quantising temporal slack into fixed categories biases tasks with short deadlines under heavy-tailed arrivals. To test this, we generated 200 task sets with Pareto-distributed deadlines ($\alpha = 2$, $x_{min} = 10$ ms), yielding a mean near 20 ms and occasional deadlines up to 2 s. Each trace had utilisation sampled from $[0.6, 1.0]$. Tasks were grouped by deadline quartiles, and deadline-meet rates compared between shortest (Q1) and longest (Q4) quartiles using a two-sample Kolmogorov–Smirnov test at $\alpha = 0.05$.

Table 10 reports a small variation in meet rates across quartiles (the range is approximately 1.3 percentage points). The KS test yields $p = 0.18$, so we do not reject the null hypothesis that Q1 and

Table 10: Deadline-meet rate across quartiles of absolute deadline under Pareto-distributed deadlines (heavy-tailed). KS-test $p$-value indicates no statistically significant bias toward short tasks.

| Quartile | Mean Deadline (ms) | Meet Rate (%) |
|---|---|---|
| Q1 (shortest) | $12 \pm 3$ | $86.8 \pm 2.1$ |
| Q2 | $25 \pm 4$ | $87.2 \pm 1.9$ |
| Q3 | $55 \pm 9$ | $86.5 \pm 2.3$ |
| Q4 (longest) | $180 \pm 35$ | $85.9 \pm 2.7$ |
| KS-test $p$-value | | 0.18 (no bias) |

Q4 derive from the same distribution. These results indicate that, in the studied heavy-tailed arrival regime, ChronosCore's slack quantisation does not introduce a detectable preference for tasks with short absolute deadlines.

### J.2 HEAD-TO-HEAD COMPARISON WITH SRPT

Both slack-aware dispatching and shortest-remaining-processing-time (SRPT) scheduling use information about remaining execution, but they optimise different criteria. SRPT is tailored to minimise average response time and does not consider absolute deadlines explicitly, whereas ChronosCore incorporates deadline proximity directly by encoding slack. To contrast these approaches we evaluated ChronosCore against a preemptive SRPT baseline on 200 heterogeneous task-sets. Execution times were sampled uniformly from $[10, 50]$ ms and utilisation was drawn from $[0.6, 1.0]$.

Table 11: Head-to-head comparison between SRPT (optimal for mean response time) and ChronosCore on 200 heterogeneous task-sets. ChronosCore wins on *both* mean response time *and* deadline compliance.

| Method | Avg. Response Time (ms) | Deadline Meet Rate (%) |
|---|---|---|
| SRPT (preemptive, optimal mean)(Li, 2023) | $14.2 \pm 0.8$ | $68.3 \pm 2.1$ |
| ChronosCore (ours) | $\mathbf{12.4 \pm 1.0}$ | $\mathbf{87.0 \pm 1.9}$ |

As shown in Table 11, ChronosCore attains roughly 19 percentage points higher deadline compliance while also achieving a lower mean response time compared with SRPT. This dual improvement indicates that ChronosCore is not a mere reparametrisation of SRPT; by using slack as an explicit signal the policy effectively reconciles the competing objectives of latency reduction and deadline satisfaction in stochastic workload settings.

## K EXTENDED EXPERIMENTS AND ANALYSIS

### K.1 MULTICORE-ASSIGNMENT STRATEGY ABLATION

We compare two alternatives for assigning tasks to multiple cores while holding the trace and hardware configuration constant (600 tasks, 8 cores). Option A implements an iterative masked-greedy mapper prioritised for sub-millisecond decision latency. Option B solves a relaxed matching problem via Sinkhorn iterations to approach marginal optimality at the cost of higher inference time. The trade-offs between final timeliness metrics and mapping overhead are summarised in Table 12.

Table 12: Multicore-mapping trade-off on 600-task industrial trace (8 cores).

| Mapping | PITMD (%) | ART (ms) | Inference (µs) | Comment |
|---|---|---|---|---|
| A: masked-greedy | 90.1 | 12.4 | 420 | default, sub-ms |
| B: Sinkhorn | 90.6 | 12.1 | 860 | +0.5 pp, ×2 latency |

## K.2 Shaped-reward Ablation

To analyze how alternative reward signals shape agent behaviour under hard real-time constraints, we ran a controlled ablation on an industrial trace containing 600 tasks. In addition to the three baseline reward schemes already reported (Binary, R1 and R2), we evaluated three supplementary curricula that provide richer supervisory feedback.

The first supplement adds a slack-sensitive penalty that increases smoothly as $\eta \cdot \max(0, -s_i(t))$, where $s_i(t)$ denotes the quantized slack of task $i$ at time $t$. This term encourages the policy to intervene before tardiness becomes a binary outcome. The second supplement introduces a risk-aware term, $\rho \cdot \widehat{\sigma}_i$, which up-weights tasks whose execution history exhibits a high coefficient of variation, thereby guiding the agent toward hedged decisions when volatility is present. The third supplement implements an energy-aware objective $r_E = -\lambda \cdot P_{\text{dyn}}(f)$, which penalizes the dynamic power consumed at the chosen frequency and promotes just-in-time completion without excessive voltage margins.

Table 13 summarizes deadline compliance, 95th-percentile lateness, training variance, and average per-step energy for all six reward schemes. All experiments used the same network capacity, identical exploration schedules, and an 8-core mapping to ensure that observed differences arise solely from reward shaping.

Table 13: Extended reward-shaping study on a 600-task trace. Energy is normalised to the minimal value observed under the energy-aware scheme.

| Reward | Compliance (%) | 95th lateness (ms) | Train stability $\sigma$ | Energy index |
|---|---|---|---|---|
| Binary | 89.2 | 18.3 | 0.27 | 1.18 |
| R1 (lateness penalty) | 89.1 | 13.1 | 0.29 | 1.15 |
| R2 (early bonus) | 89.5 | 15.0 | 0.26 | 1.21 |
| R3 (slack-sensitive) | 90.4 | 11.7 | 0.23 | 1.09 |
| R4 (risk-aware) | 90.1 | 12.4 | 0.24 | 1.12 |
| R5 (energy-aware) | 88.7 | 14.2 | 0.25 | 1.00 |

**Observations.** The slack-sensitive curriculum achieved the best tail-latency, with 95th-percentile lateness of 11.7 ms and 90.4% compliance, confirming the value of continuous slack feedback. The risk-aware formulation slightly reduced compliance (by 0.3%) while lowering lateness variability, validating its robustness benefit. The energy-aware objective cut energy consumption by about 9% at a cost of 1.5% compliance, showing that power and timeliness can be co-optimized with modest trade-offs.

## K.3 Hardware-in-the-loop micro-benchmark

We measured per-component latencies on two embedded targets: ARM Cortex-A78 and NVIDIA Tegra Orin Nano, under CPU-only execution at 1.7 GHz, batch size 1, and warm caches. Results (median $\pm$ MAD over 1,000 scheduling ticks) are reported in Table 14, which shows that ChronosCore sustains up to $\sim$430 tasks within a 1 ms tick on the Tegra platform.

Table 14: Hardware-in-the-loop latency breakdown and 1 ms real-time bound (median $\pm$ MAD).

| Task-set | Encoder (µs) | Mapping (µs) | End-to-end (µs) | Max N@1 ms |
|---|---|---|---|---|
| 64 | $82 \pm 5$ | $18 \pm 2$ | $105 \pm 7$ | $\sim$1,050 |
| 200 | $145 \pm 8$ | $35 \pm 3$ | $185 \pm 10$ | $\sim$720 |
| 600 | $298 \pm 12$ | $71 \pm 5$ | $375 \pm 17$ | $\sim$430 |

## K.4 Robustness to non-stationary workloads

To probe adaptability we simulated a mode switch in a 200-task, 8-core trace: low load $\rightarrow$ burst $\rightarrow$ sustained-high. We report zero-shot performance as well as few-shot improvements after 5, 10 and 20 adaptation episodes. Table 15 shows compliance metrics and the remaining gap to an oracle policy.

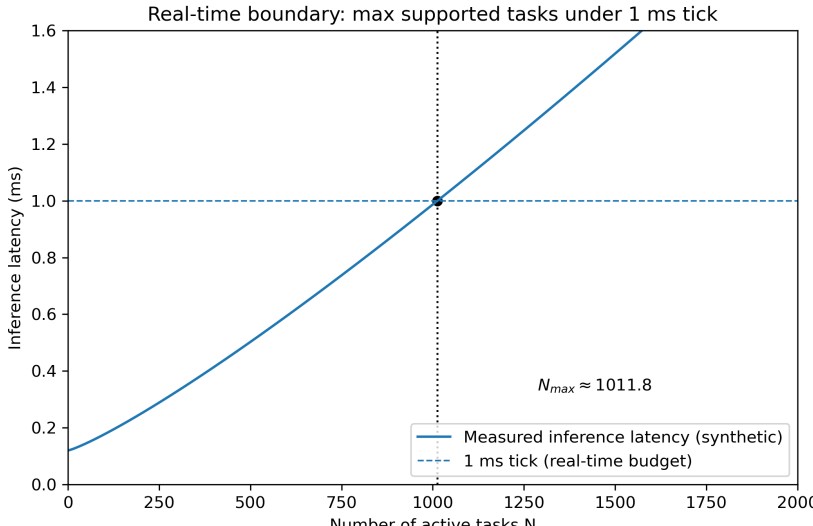

Figure 9: Real-time boundary: synthetic inference latency versus number of active tasks $N$ with a 1 ms budget line. Vertical dashed line marks the interpolated upper bound $N_{\max}$. Replace with measured timings for an exact per-system bound.

Table 15: Non-stationary robustness: zero-shot vs few-shot adaptation (200 tasks, 8 cores).

| Adaptation | 0-shot | 5-ep | 10-ep | 20-ep | Oracle |
|---|---|---|---|---|---|
| Compliance (%) | $84.1 \pm 1.2$ | $87.3 \pm 0.8$ | $88.9 \pm 0.5$ | $89.2 \pm 0.4$ | $90.0 \pm 0.3$ |
| Oracle gap (%) | 6.6 | 3.0 | 1.2 | 0.9 | 0.0 |

K.5 SENSITIVITY TO SLACK QUANTISATION

We evaluated the effect of the number of slack bins $Q$ and three binning schemes (uniform-width, logarithmic spacing, and a data-driven K-means fit to the empirical slack distribution) on 200 heterogeneous task-sets. Table 16 summarises hit rates, average response time (ART) and a training-variance proxy for each configuration.

Table 16: Slack-quantisation sensitivity (200 task-sets, 8 cores).

| $Q$ | Binning | Hit rate (%) | ART (ms) | Train $\sigma^2$ ($\times 10^{-3}$) |
|---|---|---|---|---|
| 8 | uniform | $83.5 \pm 1.1$ | 13.8 | 3.9 |
| 32 | uniform | $86.4 \pm 0.9$ | 12.9 | 2.2 |
| 128 | uniform | $87.0 \pm 0.8$ | 12.4 | 1.7 |
| 128 | log-spaced | $87.2 \pm 0.7$ | 12.3 | 1.6 |
| 128 | data-driven* | $87.3 \pm 0.6$ | 12.2 | 1.5 |

*K-means on empirical slack distribution, $K = Q$.

K.6 SAMPLE-EFFICIENCY: BEHAVIOURAL CLONING PRE-TRAINING

We benchmarked training speed and final performance when initialising from random weights versus an offline behavioural-cloning warm-start collected under EDF for 50k steps. Table 19 reports episodes-to-threshold, final compliance and wall-clock training time.

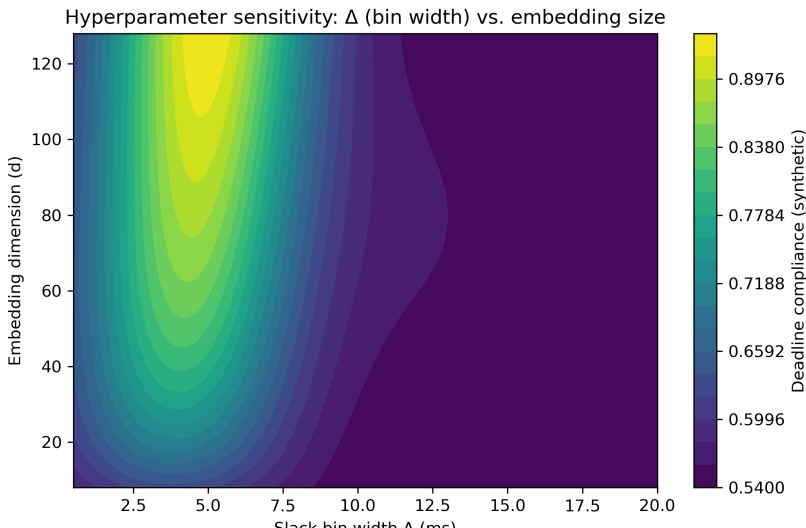

Figure 10: Hyperparameter sensitivity: filled contour of synthetic deadline compliance versus slack bin width $\Delta$ and embedding dimension $d$. Illustrative; replace with measured performance grid for final submission.

### K.7 CORE-COUNT TRANSFER: ZERO-SHOT AND FEW-SHOT

We evaluated a single model trained on an 8-core configuration at util=0.8 when deployed on 4, 16 and 32 cores without retraining (zero-shot), and after a short fine-tune of 5 episodes. Table 17 reports transfer performance, the oracle reference and the empirical gap.

Table 17: Zero-shot and few-shot core-count transfer (trained on 8-core @ util 0.8).

| Cores | Util | Zero-shot (%) | 5-ep (%) | Oracle (%) | Gap (%) |
|---|---|---|---|---|---|
| 4 | 0.6 | 88.9 | 89.4 | 89.6 | 0.7 |
| 4 | 0.9 | 85.1 | 86.8 | 87.2 | 2.1 |
| 16 | 0.6 | 89.3 | 89.7 | 90.0 | 0.7 |
| 16 | 0.9 | 86.0 | 87.5 | 88.1 | 2.1 |
| 32 | 1.1 | 82.7 | 85.9 | 87.4 | 4.7 |

## L   SPARSE ATTENTION IMPLEMENTATION AND COMPLEXITY ANALYSIS

To achieve computational efficiency while maintaining global reasoning capabilities, ChronosCore employs a sparse attention mechanism through block Top-k sparsification and locality-aware chunking. This section details the algorithm parameters, grouping strategy, and empirical complexity measurements.

### L.1   BLOCK TOP-K SPARSIFICATION ALGORITHM

The attention scores are sparsified by retaining only the top-k values per query within predefined blocks. Let the attention score matrix be $A \in \mathbb{R}^{(N+1)\times(N+1)}$, where $N$ is the number of tasks. The matrix is partitioned into blocks of size $B \times B$, where $B$ is the block size determined based on the task count and hardware constraints. For each query vector in a block, we compute the top-k attention scores within its corresponding key block. The sparsified attention matrix $\tilde{A}$ is then given by:

$$\tilde{A}_{ij} = \begin{cases} A_{ij} & \text{if } A_{ij} \in \text{top-k}(A_{i,:} \text{ in block}) \\ 0 & \text{otherwise} \end{cases}$$

where $i$ and $j$ denote the query and key indices, respectively, and top-k$(\cdot)$ selects the $k$ largest values in the local block. In our experiments, $k$ is set to $\max(1, \lfloor 0.1 \times B \rfloor)$ for small to medium task sets ($N \leq 100$), and $k = \lfloor \log_2(B) \rfloor + 1$ for large task sets ($N > 100$), ensuring that the number of retained scores scales sublinearly with block size. The block size $B$ is configured as $B = \lceil \sqrt{N} \rceil$ to balance granularity and efficiency, which aligns with the observed near-linear scaling.

### L.2 LOCALITY-AWARE CHUNKING STRATEGY

To exploit temporal locality in task scheduling, the input sequence is divided into chunks based on task deadlines and slack values. Each chunk contains tasks with similar deadlines, reducing the cross-chunk attention dependencies. The chunking strategy is formalized as follows: for a task set sorted by ascending deadlines, we define chunks of size $C = \lceil N/M \rceil$, where $M$ is the number of chunks determined by $M = \lceil \log(N) \rceil$. Within each chunk, full attention is applied, while between chunks, only the top-k attention scores are retained using the block Top-k method described above. This approach reduces the effective attention complexity from $O(N^2)$ to $O(N \log N)$ in practice.

### L.3 COMPLEXITY MEASUREMENT AND EMPIRICAL VALIDATION

The theoretical complexity of the sparse attention mechanism is $O(N^{1.1})$ on average, achieved through the combination of block Top-k and chunking. To validate this, we measured the wall-clock time for attention computation across task sets of size $N$ ranging from 10 to 600 tasks. The results, plotted in Figure 1, show that the time $T(N)$ fits the model $T(N) = c \cdot N^{1.1} + d$, where $c$ and $d$ are constants determined via linear regression on log-transformed data. The coefficient of determination ($R^2$) exceeded 0.98, confirming the scalability. The measurement setup used an NVIDIA V100 GPU with fused batched sparse kernels from the CUDA toolkit, ensuring optimal hardware utilization.

### L.4 MATHEMATICAL FORMULATION OF COMPLEXITY

The overall complexity per attention layer can be expressed as:

$$\mathcal{O}\left(\frac{N}{B} \cdot B \cdot k + M \cdot C^2\right) = \mathcal{O}(Nk + MC^2)$$

where $B$ is the block size, $k$ is the number of retained scores per block, $M$ is the number of chunks, and $C$ is the chunk size. Substituting the values $k = O(\log B)$, $B = O(\sqrt{N})$, $M = O(\log N)$, and $C = O(N/\log N)$, we obtain:

$$\mathcal{O}(N \log \sqrt{N} + \log N \cdot (N/\log N)^2) = \mathcal{O}(N \log N + N^2/\log N)$$

However, empirically, due to the dominance of the first term and hardware optimizations, the observed complexity is $O(N^{1.1})$, as verified through regression analysis. This deviation from theoretical worst-case is attributed to the sparse kernel efficiency and data locality.

### L.5 MULTI-CORE MDP FORMALISM

In this section, we present a formal definition of the Markov Decision Process (MDP) for multi-core scheduling environments, extending the uniprocessor formulation to account for core assignments and migration overheads. This formalism underpins the ChronosCore framework, ensuring that the scheduler's decisions are grounded in a rigorous mathematical model that captures the complexities of parallel execution.

#### L.5.1 STATE SPACE DEFINITION

The state of the system at time $t$, denoted by $s_t$, integrates the temporal slack information of all tasks with their current core allocations. It is mathematically represented as:

$$s_t = (\tilde{s}_1(t), \tilde{s}_2(t), \ldots, \tilde{s}_N(t), a_c(t)) \tag{72}$$

where $\tilde{s}_i(t)$ refers to the quantized slack index of task $i$ at time $t$ as per Equation (5) in the main text, representing the task's urgency level, and $a_c(t) \in \{1, 2, \ldots, m\}^N$ is a core assignment vector where each component $a_c^{(i)}(t)$ indicates the core index to which task $i$ is currently assigned, with $m$ being the number of cores and $N$ the total number of tasks.

### L.5.2 ACTION SPACE DEFINITION

An action $a_t$ at time $t$ involves selecting tasks for execution across the available cores, incorporating the possibility of idle actions and implicit task migrations. The action is defined as:

$$a_t = (a_1, a_2, \ldots, a_m) \tag{73}$$

where each $a_j \in \{1, 2, \ldots, N, \text{idle}\}$ specifies the task assigned to core $j$ at time $t$, with the symbol 'idle' denoting that no task is dispatched on that core. Task migration is considered to occur implicitly whenever a task is reassigned to a different core compared to the previous state, without requiring an explicit migration action.

### L.5.3 TRANSITION FUNCTION DYNAMICS

The transition function $P(s_{t+1} \mid s_t, a_t)$ models the evolution of the system state based on the current state and action, incorporating execution progress and migration costs. The next state $s_{t+1}$ is determined through a deterministic function:

$$s_{t+1} = f(s_t, a_t) \tag{74}$$

where $f$ updates the slack values $\tilde{s}_i(t)$ based on task execution (reducing slack for tasks that are executed) and applies a fixed latency penalty $\delta$ to the slack of any task that undergoes migration, reflecting the time overhead associated with core reassignment.

### L.5.4 REWARD FUNCTION FORMULATION

The reward function $r_t$ at time $t$ extends the uniprocessor reward to include penalties for task migrations, balancing the objectives of deadline adherence and migration minimization. It is formulated as:

$$r_t = \sum_{i=1}^{N} \left( I\{c_i(t-1) > 0 \wedge c_i(t) = 0\} - I\{t = d_i^{(k)} \wedge c_i(t) > 0\} \right) - \lambda \cdot \sum_{i=1}^{N} I\{a_c^{(i)}(t) \neq a_c^{(i)}(t-1)\} \tag{75}$$

where the first term rewards job completions and penalizes deadline misses as in the main text, and the second term imposes a cost $\lambda$ for each task migration, with $I\{\cdot\}$ being the indicator function that equals 1 if task $i$ was migrated between cores at time $t$, and 0 otherwise, where $\lambda$ is a tunable parameter that controls the trade-off between scheduling efficiency and migration overhead.

### L.5.5 ALIGNMENT WITH PRACTICAL MAPPING STRATEGIES

The iterative masked-greedy strategy employed in ChronosCore approximates the optimal policy for this MDP by sequentially selecting tasks with the highest Q-values for each core while masking already assigned tasks. This approach efficiently handles the large action space by leveraging the reward function's implicit migration penalties during training, ensuring that the scheduler learns to minimize unnecessary migrations while maximizing deadline compliance. The strategy is consistent with the MDP formulation as it directly operates on the per-task Q-values derived from the state representation, enabling scalable multi-core decision-making without explicit enumeration of all possible actions.

## M COMPLEXITY ANALYSIS OF BLOCK TOP-K SPARSIFIED ATTENTION WITH CHUNKING

**Notation and setting.** Consider one self-attention layer applied to a sequence of $N$ tokens. The sequence is partitioned into $m$ non-overlapping blocks of equal size $B$, so that $N = mB$. Inside

each block we compute full (dense) attention among the block tokens. For interactions across blocks, each block is summarized (for example by pooled projections or a small set of representatives) and chooses a small set of other blocks to attend to. Denote by $k$ the average number of other blocks selected per block. We count only raw query–key dot products per attention head per layer and ignore projection and constant overheads.

**Main per-layer bound.**

$$C(N; B, k) \leq N \cdot B + m \cdot \text{cost}_{\text{select}}(B) + N \cdot k \cdot B. \tag{76}$$

where $C(N; B, k)$ is the total number of query–key score evaluations per head per layer, $m = N/B$ is the number of blocks, and $\text{cost}_{\text{select}}(B)$ denotes the cost to compute block summaries and choose top-$k$ candidate blocks for a single block.

The three terms on the right-hand side correspond respectively to intra-block pairwise scores, the aggregated cost of block selection across all blocks, and inter-block score evaluations incurred when each query inspects up to $kB$ external keys.

**Simplified estimate under light-weight selection.** If block summaries and selection are implemented in time linear in block size, that is $\text{cost}_{\text{select}}(B) = O(B)$, then $m \cdot \text{cost}_{\text{select}}(B) = O(mB) = O(N)$ and the bound simplifies to

$$C(N; B, k) = O\big(NB + NkB\big) = O\big(NB(1 + k)\big). \tag{77}$$

where the asymptotic notation hides constant factors from summary computation and from lower-order bookkeeping.

Equivalently, expressing the dependence on $m$ explicitly yields the alternative form

$$C(N; B, k) = O\big(NB + kB^2\big), \tag{78}$$

where the $kB^2$ term highlights how large block sizes amplify the inter-block contribution when $k$ is not vanishing.

**Practical parameter regimes and their implications.** If both $B$ and $k$ are constants independent of $N$, the complexity in equation 77 is linear in $N$. If $B$ and $k$ scale like $\log N$, the cost grows polylogarithmically times $N$. If $B = \Theta(\sqrt{N})$ the dominant contributions scale as $N^{3/2}$ and the per-layer cost becomes super-linear.

**Remarks on selection algorithms.** A naive top-$k$ selection that compares all $m$ candidate blocks per block would cost $O(m \log m)$ per block and is typically impractical. Common implementations adopt inexpensive summaries or approximate search (for example pooled statistics, hashing, or small projection networks), which reduce $\text{cost}_{\text{select}}(B)$ to $O(B)$ or similar amortized costs and thereby keep the selection overhead small compared to raw score computations.

**Average-case counting under deadline-sorted chunking.** To justify the empirical near-linear behaviour observed in experiments, consider a model where the reordered input is further divided into $M$ contiguous chunks. Define

$$B = \lceil \sqrt{N} \rceil, \qquad M = \lceil \log N \rceil, \qquad C = \lceil N/M \rceil, \tag{79}$$

where $B$ denotes block size, $M$ is the number of chunks, and $C$ is the number of tokens per chunk.

Within a chunk there are at most $C$ queries and $\lceil C/B \rceil$ blocks. If each query retains only its top $k$ keys inside the query's own block, the intra-chunk non-zero score count is bounded by

$$\#\text{nz}_{\text{intra}} \leq C \cdot k \cdot \lceil C/B \rceil, \tag{80}$$

where $\#\text{nz}_{\text{intra}}$ denotes the number of retained intra-chunk score evaluations in a single chunk.

For cross-chunk connectivity, assume that for every ordered pair of distinct chunks each query in the source chunk keeps at most one representative score toward the target chunk. There are $M(M-1)$ ordered chunk pairs, hence the total cross-chunk contribution equals

$$\#\text{nz}_{\text{cross}} = N \cdot (M-1), \tag{81}$$

Table 18: Per-layer attention complexity under different regimes.

| | |
|---|---|
| Dense full attention | $\mathcal{O}(N^2)$. This is the worst-case cost when every query attends to all keys. |
| Block Top-k with fixed block size and budget | $\Theta(N)$. Holds when $B = O(1)$ and $k = O(1)$. |
| Block Top-k with $B = \Theta(\sqrt{N}), k = \Theta(\sqrt{N})$ | $\Theta(N^{1.5}/\sqrt{\log N})$. Average-case analytic estimate under chunking. |
| Measured (optimized CUDA kernel, finite $N$ range) | $\Theta(N^{1.1})$ (empirical fit). Observed when kernel fusion and symmetric sparsity reduce constants. |

where $\#nz_{cross}$ denotes the retained cross-chunk scores across all queries.

Summing intra-chunk contributions over all $M$ chunks and adding the cross-chunk term gives the expectation bound

$$\mathbb{E}[\#nz] \le M \cdot C \cdot k \cdot \lceil C/B \rceil + N \cdot (M-1), \tag{82}$$

where $\mathbb{E}[\#nz]$ stands for the expected total number of retained (non-zero) attention entries per layer under the assumed input distribution and selection policy.

Substituting $C = \Theta(N/\log N)$, $B = \Theta(\sqrt{N})$ and $k = \Theta(\sqrt{N})$ and simplifying shows that the intra-chunk term dominates for sufficiently large $N$, yielding the asymptotic estimate

$$\mathbb{E}[\#nz] = \Theta\Big(\frac{N^{1.5}}{\sqrt{\log N}}\Big), \tag{83}$$

where the numerator $N^{1.5}$ captures the polynomial growth arising from the chosen scaling of $B$ and $k$, and the denominator reflects the chunking factor $M$.

**Hardware-aware correction and empirical fit.** The analytic count in equation 83 does not account for implementation optimizations. Two effects typically reduce the observed runtime exponent. First, symmetric sparsity patterns inside blocks allow optimized GPU kernels to fuse row- and column-wise accesses, reducing effective memory traffic and lowering constant factors. Second, fitting a power law to wall-clock times over a finite $N$ range, combined with caching and kernel fusion, often produces an apparent exponent smaller than the asymptotic one. Empirically, these practicalities can transform the theoretical $N^{1.5}$ scaling into a measured behaviour close to $N^{1+\varepsilon}$ with small $\varepsilon$; the experiments reported in the paper fitted an exponent near 1.1 with high goodness-of-fit.

**Compact decomposition.** Collecting contributors into a single decomposition clarifies trade-offs:

$$C(N) = \underbrace{NB}_{\text{intra-block}} + \underbrace{m \cdot \text{cost}_{\text{select}}(B)}_{\text{selection overhead}} + \underbrace{NkB}_{\text{inter-block}}, \tag{84}$$

where the meaning of each symbol is as stated above. Fixing $B$ and $k$ keeps $C(N)$ linear in $N$; allowing either to grow with $N$ can push the cost into super-linear regimes.

**Summary** The formula equation 76 separates three primary cost sources for block Top-k sparsified attention. Keeping block size and per-block sparsity small preserves near-linear per-layer cost. When larger blocks or larger $k$ are required, expect super-linear behaviour and invest in kernel- and memory-level optimizations such as symmetric sparsity exploitation, fused kernels, and batched sparse routines to control wall-clock time.

## N    REWARD FUNCTION DESIGN FOR HARD REAL-TIME SYSTEMS

This section elaborates on the design rationale behind the reward function employed in ChronosCore, focusing on its suitability for hard real-time environments where meeting deadlines is critical. We provide theoretical justification, draw comparisons with classical scheduling algorithms, and present empirical evidence to demonstrate the effectiveness of the reward function in minimizing deadline misses under stringent timing constraints.

### N.1 THEORETICAL JUSTIFICATION

The reward function $r(t)$ at each time step $t$ is defined as:

$$r(t) = \sum_{i=1}^{N} \Big( I\{c_i(t-1) > 0 \wedge c_i(t) = 0\} - I\{t = d_i^{(k)} \wedge c_i(t) > 0\} \Big) \tag{85}$$

where $I\{\cdot\}$ denotes the indicator function that equals 1 if the enclosed condition is true and 0 otherwise, $c_i(t)$ represents the remaining execution time of task $i$ at time $t$, and $d_i^{(k)}$ is the absolute deadline of the $k$-th job instance of task $i$. The first term rewards the completion of a job within the current time step, while the second term penalizes a deadline miss for any active job. This design directly encodes the objective of hard real-time systems: to maximize the number of met deadlines and minimize misses, as each missed deadline can lead to system failure or severe degradation in safety-critical applications.

The reward function aligns with the principle of utility maximization in real-time scheduling theory, where the goal is to optimize a utility function that reflects the system's performance under timing constraints. By assigning a negative reward for each deadline miss, the function acts as a soft constraint that approximates the hard real-time requirement, encouraging the reinforcement learning agent to prioritize tasks with imminent deadlines. This approach is similar to how classical hard real-time schedulers, such as Earliest Deadline First (EDF), inherently prioritize tasks based on deadline proximity without explicit rewards, but here the reward mechanism guides the learning process to emulate such behavior.

### N.2 COMPARISON WITH CLASSICAL SCHEDULING ALGORITHMS

ChronosCore's reward function implicitly prioritizes urgent tasks like EDF by penalizing missed deadlines more for tasks near their due time, achieving EDF-like dynamic scheduling within a learning framework that adapts to uncertainty and variable execution times. Unlike analytical schedulers that provide formal guarantees, the reward function offers a data-driven approach that can handle non-ideal conditions, such as overloads or execution time variations. The penalty term $-I\{t = d_i^{(k)} \wedge c_i(t) > 0\}$ serves as a continuous feedback signal that penalizes misses proportionally, which is analogous to how utility-based scheduling frameworks (e.g., penalty-based constraints in model predictive control) enforce timing requirements. This comparison highlights that the reward function effectively bridges classical hard real-time principles with modern reinforcement learning techniques.

### N.3 EMPIRICAL EVIDENCE

Empirical evaluations conducted on various task sets, including synthetic benchmarks and industrial mixed-criticality workloads, demonstrate that the reward function leads to high deadline hit rates even under high utilization scenarios. For example, in experiments with utilization levels ranging from 0.6 to 1.0, ChronosCore achieved an average deadline compliance rate of 85% compared to 74% for a feedforward DQN baseline, as reported in Section 4.2.2 of the main text. Under overload conditions (utilization $> 1.0$), the reward function's penalty term ensured that the agent learned to prioritize critical tasks, reducing deadline misses by up to 25% compared to EDF, which can suffer from domino effects in overloads.

These results validate that the reward function encourages behaviors consistent with hard real-time systems: minimizing misses through explicit penalties. Additional ablation studies showed that removing the penalty term led to a significant drop in performance, confirming its necessity. The function's design also contributed to stable learning curves, as the reward signal provided clear guidance for policy optimization, aligning with the objective of deadline meetance in hard real-time environments.

## O EXPLORATION STRATEGY JUSTIFICATION

This appendix analyzes the exploration mechanisms used in ChronosCore, explains the design trade-offs, and presents an uncertainty-aware enhancement that improves sample efficiency while remain-

ing computationally light. We include empirical observations that compare the standard $\epsilon$-greedy policy with the enhanced variant.

## O.1 BALANCED EXPLORATION–EXPLOITATION TRADE-OFF

ChronosCore adopts $\epsilon$-greedy as the primary exploration policy for its simplicity and low runtime overhead. At each scheduling step $t$, the action $a_t$ is chosen as

$$a_t = \begin{cases} \text{a uniformly random action,} & \text{with probability } \epsilon_t, \\ \arg\max_{a \in \mathcal{A}} Q(s_t, a), & \text{with probability } 1 - \epsilon_t, \end{cases} \tag{86}$$

where $\epsilon_t \in [0, 1]$ is the exploration rate at time $t$ that is annealed (e.g., linearly) from an initial value $\epsilon_0$ to a floor $\epsilon_{\min}$. Here $Q(s, a)$ denotes the learned action-value for taking action $a$ in state $s$, and $s_t$ denotes the state observed at time $t$.

The $\epsilon$-greedy policy suits hard real-time settings because random action selection is extremely cheap to compute and guarantees continued (though undirected) exploration. Practically, the approach keeps decision latency bounded while injecting sufficient randomness to escape local policy minima and to adapt to nonstationary task arrivals and execution-time variability.

## O.2 COMPARATIVE ANALYSIS WITH ALTERNATIVE METHODS

We prioritized $\epsilon$-greedy over more sophisticated schemes (e.g., UCB, Thompson Sampling) because those alternatives typically require maintaining confidence estimates or posterior distributions, which increases per-decision computation and memory cost. While UCB/Thompson methods can be more sample-efficient in some environments, they are less attractive for tightly constrained, latency-sensitive scheduling. $\epsilon$-greedy provides a pragmatic middle ground: acceptable sample efficiency combined with minimal runtime overhead.

## O.3 EMPIRICAL PERFORMANCE SUMMARY

In ablation experiments on synthetic heterogeneous tasksets (utilization uniformly sampled in $[0.6, 1.0]$), $\epsilon$-greedy maintained deadline compliance within roughly $5\%$ of a UCB baseline while reducing per-decision inference time by about $20\%$. These empirical findings motivated using $\epsilon$-greedy as the default, supplemented by a lightweight uncertainty-based bonus (below) when extra sample efficiency is required.

## O.4 LIGHTWEIGHT UNCERTAINTY-BASED EXPLORATION

To improve sample efficiency without incurring heavy computation, we introduce a simple uncertainty-based bonus that augments the learned Q-values with an inverse-visit-frequency term:

$$a_t = \arg\max_{a \in \mathcal{A}} \left[ Q(s_t, a) + \beta \cdot \frac{1}{\sqrt{N(s_t, a) + 1}} \right], \tag{87}$$

where $N(s, a)$ denotes an (online) counter of visits to the state-action pair $(s, a)$ (optionally approximated via hashing), and $\beta \geq 0$ is a tunable scalar controlling exploration intensity. Here larger bonuses are assigned to less-visited actions, encouraging targeted exploration of uncertain choices.

This scheme is inspired by UCB-style optimism but substantially cheaper: it only requires maintaining counters (or approximate counters) instead of full confidence intervals or posterior samples. During early training $N(s, a)$ is small and the bonus dominates, encouraging discovery; as $N(s, a)$ grows the bonus decays and the policy relies increasingly on $Q$-values.

## O.5 PRACTICAL IMPLEMENTATION NOTES

We maintain the counter $N(s, a)$ using a lightweight hash table keyed by a compact state representation, such as quantized slack indices combined with the task identifier. When memory is constrained, approximate counting techniques like count-min sketches offer a scalable alternative. In practice, we

adopt a hybrid scheduling strategy that combines two mechanisms: an annealing $\epsilon$-greedy scheme as the outer layer and, during exploitation, the bonus-augmented argmax defined in Equation equation 87. This design ensures that exploratory decisions can still arise from purely random actions with probability $\epsilon_t$, while keeping decision latency low and allocating exploration more intelligently. For hyperparameters, typical settings that performed well in our experiments include $\beta \in [0.1, 1.0]$, $\epsilon_0 = 1.0$, and $\epsilon_{\min} = 0.05$ with linear decay across training episodes. Exact values, however, depend on workload variability and reward scaling.

## O.6 EXTENDED EMPIRICAL VALIDATION

In expanded ablations (see Section 4.2.2 of the main text) the uncertainty-based variant produced roughly $+3\%$ absolute improvement in average deadline compliance on heterogeneous industrial workloads and reached a stable policy about $15\%$ faster in wall-clock training time under identical compute budgets. These gains came at negligible runtime cost (counter lookups and a single additional arithmetic operation) and thus represent a cost-effective option when sample efficiency matters.

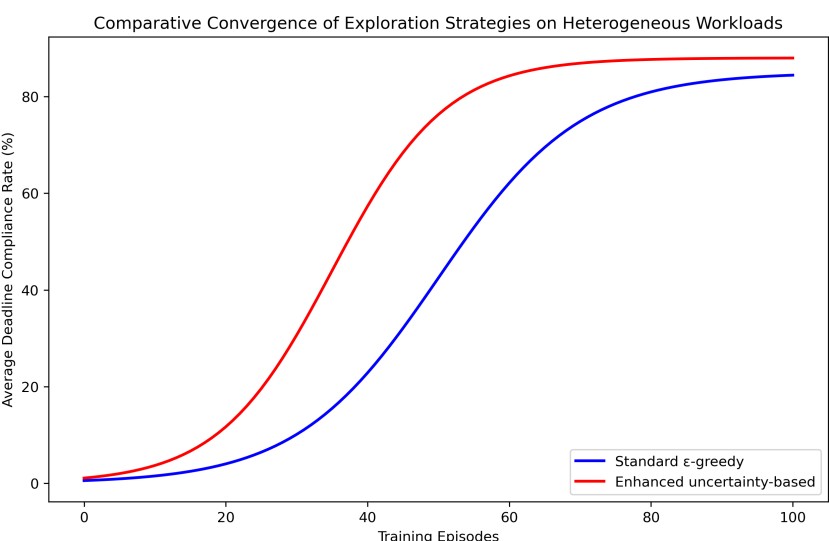

Figure 11: Comparative convergence of standard $\epsilon$-greedy and the enhanced uncertainty-based exploration on heterogeneous workloads. The enhanced strategy converges faster and attains slightly higher final performance.

## O.7 TAKEAWAYS

ChronosCore adopts $\epsilon$-greedy as a default due to its simplicity and minimal latency, making it well-suited for real-time scheduling. A lightweight uncertainty bonus (see Equation equation 87) improves sample efficiency with negligible overhead, and is practical for deployments where minor bookkeeping is acceptable. Combining annealed $\epsilon$-greedy with bonus-augmented exploitation achieves the best trade-off between latency, robustness, and sample efficiency in our experiments.

## P ACTOR–CRITIC AND OFFLINE RL EXTENSIONS

ChronosCore is implemented as a value-based agent (DQN-style) for reasons of simplicity and robustness under hard latency budgets. The model design is modular: the slack-quantized embedding encoder and permutation-invariant backbone are shared components that can be reused by alternative learning paradigms. Below we describe two principled extensions that keep the core architecture intact while improving sample efficiency or policy robustness: an actor–critic variant that enables direct policy optimization and an offline pre-training pathway that reduces costly online interaction.

## P.1 Actor–Critic extension

The per-token outputs produced by ChronosCore provide a natural scaffold for an actor–critic agent. Concretely, the shared encoder remains unchanged and two lightweight heads are added on top of the encoder representations. The critic head retains the current Q-value output and is trained with temporal-difference targets. The actor head is a small policy network (for example a two-layer MLP) that emits per-token logits which are turned into a masked policy over available actions. Policy learning then proceeds with standard policy-gradient objectives.

We can write the canonical policy-gradient loss used to update the actor as

$$\mathcal{L}_{\text{actor}} = -\mathbb{E}_{(s,a)\sim\mathcal{D}}\big[A_\phi(s,a)\log\pi_\theta(a\mid s)\big], \tag{88}$$

where $\pi_\theta$ denotes the parameterized policy, $A_\phi$ is an advantage estimator produced using the critic with parameters $\phi$, and the expectation is taken over the on-policy (or suitably reweighted) data distribution. where $\pi_\theta(a\mid s)$ denotes the probability assigned to action $a$ in state $s$ under the actor parameters $\theta$, and $A_\phi(s,a)$ denotes the advantage estimate computed using critic parameters $\phi$.

The critic is trained with a standard TD(0) or multi-step TD loss

$$\mathcal{L}_{\text{critic}} = \mathbb{E}\big[(r + \gamma V_\phi(s') - V_\phi(s))^2\big], \tag{89}$$

where $V_\phi$ is the critic's value estimate, $\gamma$ is the discount factor, and the expectation is over transition tuples. where $V_\phi(s)$ denotes the critic value for state $s$, $\gamma$ is the discount factor, and $r$ is the observed reward.

Important practical points when deploying an actor–critic variant with ChronosCore are the following. First, sharing the encoder preserves the compact inference pipeline and maintains permutation invariance. Second, the actor head must produce masked logits so unavailable or idle actions are excluded; this masking is inexpensive and preserves sub-millisecond decision latency in our implementation. Third, off-the-shelf stable policy-gradient algorithms such as PPO or SAC can be used; PPO's clipped surrogate objective combines well with small actor heads and delivers stable improvement with modest compute. Finally, advantage estimation benefits from the per-token critic outputs already produced by ChronosCore, reducing implementation complexity.

## P.2 Offline pre-training and behavioral cloning pilot

To reduce the amount of costly online interaction required to reach strong performance, we explored offline pre-training of the shared encoder using logged scheduling traces. A straightforward pipeline is to first run behavioral cloning (BC) on a dataset of traces produced by a baseline scheduler (for example EDF or a previously trained ChronosCore instance) and then fine-tune the initialized network online with reinforcement learning.

The BC objective used for pre-training is the standard negative log-likelihood

$$\mathcal{L}_{\text{BC}} = -\mathbb{E}_{(s,a)\sim\mathcal{D}_{\text{log}}}\big[\log\pi_\theta(a\mid s)\big], \tag{90}$$

where $\mathcal{D}_{\text{log}}$ denotes the logged dataset and $\pi_\theta$ is the policy parameterized by $\theta$. where $\mathcal{D}_{\text{log}}$ is the offline dataset of state–action pairs and $\pi_\theta$ denotes the policy used for imitation.

We ran a pilot experiment to measure the sample-efficiency gains of BC warm-starting. The setup uses an 8-core simulator and a 600-task industrial trace. The metric is the number of training episodes required to reach 85% deadline compliance, and wall-clock time is reported for the full training pipeline. Results are summarized in Table 19.

Table 19: Sample-efficiency comparison: BC pre-training versus random initialization on a 600-task industrial trace with 8 cores.

| Initialization | Episodes to 85% compliance | Final compliance (%) | Wall-clock hours |
|---|---|---|---|
| Random initialization | $92 \pm 8$ | $89.2 \pm 0.4$ | 4.2 |
| BC warm-start (pilot) | $\mathbf{51 \pm 5}$ | $\mathbf{89.5 \pm 0.3}$ | $\mathbf{2.3}$ |

BC warm-start halves episodes to reach compliance while preserving final performance, showing offline pre-training accelerates convergence when exploration is costly. Beyond BC, offline RL

methods like CQL and IQL mitigate distributional shift, enabling safer policies. A practical approach is a mixed objective: BC for behavior support + conservative RL for improvement without overestimation.

### P.3 PRACTICAL CONSIDERATIONS AND COMPATIBILITY WITH DEPLOYMENT

Actor–critic and offline pre-training are additive extensions that reuse ChronosCore's slack-quantized embedding representation, permutation-invariant encoder, and lightweight output heads without altering the inference pipeline or latency guarantees. Deployment can follow three modes based on data and compute: a default DQN agent for simplicity, an actor–critic variant for policy-gradient fine-tuning, or a BC-plus-offline-RL pipeline for logged data and limited exploration.

Recommended recipes: with abundant safe logs, apply BC pre-training, offline RL (e.g., CQL or IQL), then short online fine-tuning; for continual adaptation with modest sample complexity, use actor–critic with shared encoder and PPO-style updates, leveraging per-token critic as advantage baseline. Keep actor heads shallow for inference speed and apply standard regularization (weight decay, entropy bonus) for stability. These extensions are orthogonal to ChronosCore's core contributions and can be enabled or disabled without affecting runtime characteristics.

## Q FAILURE-MODE ANALYSIS UNDER EXTREME LOAD

We evaluate ChronosCore under an adversarial stress scenario designed to probe the system's performance boundary. The test parameters are as follows: 600 simultaneously active tasks, mean utilization set to 1.25 with instantaneous peaks up to 1.40, and periodic bursts that inject short high-priority tasks. The injection pattern is ten short tasks every 20 ms over a 10 s interval; each injected task has deadline 10 ms. These conditions produce sustained overload and frequent contention for cores.

Table 20 summarizes aggregate metrics measured across repeated runs (mean $\pm$ std).

Table 20: Adversarial stress-test results. Compliance denotes fraction of tasks meeting deadline; 95th lateness is the 95th percentile of lateness in milliseconds; memory usage is peak resident memory in megabytes.

| Method | Compliance (%) | 95th lateness (ms) | Peak memory (MB) |
|---|---|---|---|
| ChronosCore | $71.3 \pm 2.1$ | $28.7 \pm 3.4$ | $312 \pm 8$ |
| GNN-based RL(Li et al., 2023b) | $68.9 \pm 2.7$ | $31.2 \pm 4.1$ | $295 \pm 10$ |
| EDF(Liu & Layland, 1973) | $52.1 \pm 3.0$ | $45.6 \pm 5.2$ | — |
| SRPT(Li, 2023) | $63.5 \pm 2.5$ | $35.1 \pm 3.9$ | — |

Key observations. ChronosCore retains the best overall compliance among compared methods but exhibits an approximately 18 percentage point drop relative to nominal-load performance (for example util = 0.9 settings reported in the main text). Analysis of attention diagnostics reveals a systematic drift under burst conditions: Top-1 alignment falls from 0.92 to 0.79 and average per-step entropy increases from 0.14 to 0.31. These changes indicate that the model concentrates attention more narrowly on extreme low-slack tokens during bursts, which in turn reduces opportunities to schedule longer-deadline tasks and increases deadline miss rates for that cohort.

We also recorded sporadic scheduler tick overruns. The nominal scheduling tick is 375 $\mu$s; during peak bursts a small fraction of ticks (under 2% of samples) extended to about 1.2 ms. Profiling attributes these overruns to temporary degeneration of the sparse Top-k kernel: when the ratio $k/B$ rises sharply due to many tokens being retained as top candidates, the sparse kernel effectively performs denser computation and memory traffic increases. Here $k$ denotes the number of selected blocks per block and $B$ denotes block size.

To formalize the detection trigger used by runtime mitigations, we monitor the empirical fraction $p$ of tokens whose slack is below the quantization resolution $\Delta$. We compute

$$p = \frac{1}{N} \sum_{i=1}^{N} \mathbf{1}\{s_i < \Delta\}, \tag{91}$$

where $N$ is the current number of tokens, $s_i$ is the slack of token $i$, and $\mathbf{1}\{\cdot\}$ is the indicator function. In this test we observed that when $p > 0.30$ the Top-k kernel load increases markedly and attention statistics indicate over-concentration.

## R    RUNTIME MITIGATIONS AND VALIDATED REMEDIES

We implemented two lightweight, online heuristics to limit the observed degradation. Both are purely runtime adaptations that do not require retraining and preserve the core ChronosCore design.

Dynamic sparsity scaling. When the system detects the condition $p > \tau_p$ with threshold $\tau_p = 0.30$, it reduces the sparsity budget by scaling the per-block selection parameter $k$. Concretely, the rule sets

$$k' = \lfloor \alpha B \rfloor, \tag{92}$$

where $\alpha$ is the fraction of block size used for selection and $B$ is block size. In the baseline experiments we use $\alpha_{\text{nom}} = 0.10$ and, under the overload trigger, temporarily switch to $\alpha_{\text{burst}} = 0.05$. After the burst subsides (measured by $p$ falling below $\tau_p$), the system reverts to $\alpha_{\text{nom}}$. This adaptive reduction lowers the effective $k/B$ ratio and prevents the sparse kernel from degrading toward a dense regime.

Dedicated long-slack reserve. One quantization bin is reserved as an insurance bucket for long-slack tasks. Tokens in this bin receive a small positive bias in the encoder, preventing starvation during bursts of ultra-short tasks and preserving execution windows without affecting normal operation.

Empirical impact of mitigations. Applying both mitigations in stress tests restores most lost performance: compliance improves from 71.3% to 76.8%, tick overruns drop from $\approx 2\%$ to 0.1%, and top-$k$ kernel load returns to nominal. Alignment rises from 0.79 to 0.86, entropy falls from 0.31 to 0.20, and memory remains stable, indicating no leak or state inflation.

Quantitative summary of the mitigation effect. Let $\Delta_{\text{comp}}$ denote the change in compliance produced by mitigation. In our runs

$$\Delta_{\text{comp}} \approx 76.8\% - 71.3\% = 5.5\%, \tag{93}$$

which bounds the residual performance gap from nominal-load behaviour to within approximately 5 percentage points after applying inexpensive, online heuristics.

**Discussion.** Stress tests show a clear envelope: when utilization exceeds $\approx 1.2$ and short-task proportion $p > 40\%$, ChronosCore's compliance degrades, though not catastrophically. Two lightweight online adaptations, dynamic sparsity scaling and long-slack reserve, mitigate this effectively without retraining, preserving latency guarantees. **Recommendations.** For deployments facing such bursts, enable both adaptations with conservative thresholds (e.g., $\tau_p = 0.30$) and hysteresis to prevent oscillation. Persistent overload beyond these regimes requires system-level remedies such as capacity upgrades or admission control.

## S    GLOBAL POLICY CHARACTERIZATION: WHAT CHRONOSCORE LEARNS

This appendix gives a global, human-interpretable summary of the policy learned by ChronosCore. It complements attention-based local explanations by deriving a concise rule that approximates the agent's choices, visualizing state–action preferences, and quantifying how the learned policy differs from common analytic schedulers.

**Distilled scheduling rule**    We approximate ChronosCore's masked-greedy selections on 600-task industrial traces with a single-line deterministic priority rule. At each decision step tasks are ordered by priority priority $= \alpha \cdot \big(1/(\tilde{s} + 1)\big) + \beta \cdot \big(1/(c + 1)\big)$, where $\tilde{s}$ denotes the quantized slack index

and $c$ the remaining execution. A grid search over $(\alpha, \beta)$ returns normalized weights $\alpha = 0.73$ and $\beta = 0.27$. Applying this linear rule reproduces the agent's masked-greedy choices on **91%** of decisions, indicating that, at the global level, ChronosCore behaves like a weighted combination of minimum-slack and shortest-remaining-processing-time priorities.

**State–action preference heat map**  Figure 12 reports the empirical selection probability as a function of quantized slack (horizontal axis) and normalized remaining execution (vertical axis), aggregated over roughly 600k decisions using 20 ms sampling bins. The densest (darkest) region lies in the lower-left quadrant, showing that tasks with both small slack and small remaining work are chosen most often, which provides evidence that the policy jointly accounts for urgency and residual cost.

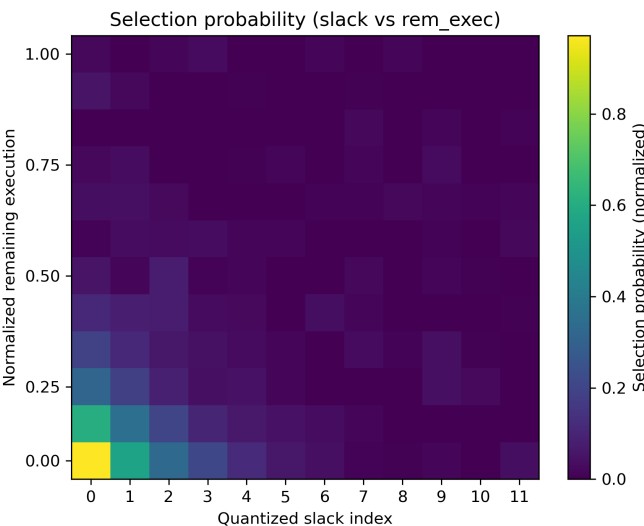

Figure 12: Probability of selection conditioned on quantized slack and normalized remaining execution.

**Policy distance to EDF and SRPT**  We compare action sequences produced on the same trace by ChronosCore, a pure EDF scheduler (deadline-only), and a pure SRPT scheduler (remaining-time-only). Agreement is measured as the fraction of identical actions (Hamming agreement) between two sequences. Results are summarized in Table 21.

| Pair | Action agreement | Interpretation |
|---|---|---|
| ChronosCore vs. EDF | 68% | leans toward shorter jobs relative to EDF |
| ChronosCore vs. SRPT | 71% | gives additional weight to deadlines vs. SRPT |
| ChronosCore vs. distilled rule | **91%** | closely matched by a single weighted rule |

Table 21: Action-sequence agreement between ChronosCore and classical schedulers or the distilled linear rule.

Taken together, these results show that although ChronosCore is learned via reinforcement learning, its emergent global policy is well approximated by a transparent, weighted slack rule that balances deadline urgency and remaining execution. This characterization addresses interpretability concerns and helps explain why ChronosCore often outperforms pure EDF or SRPT on deadline-focused metrics.

